# Protein-protein interaction network controlling establishment and maintenance of switchable cell polarity

**Luís António Menezes Carreira**[1] ⊙, **Filipe Tostevin**[1,2] ⊙, **Ulrich Gerland**[2], **Lotte Søgaard-Andersen**[1] *

1 Department of Ecophysiology, Max Planck Institute for Terrestrial Microbiology, Marburg, Germany,
2 Physik-Department, Technische Universität München, James Franck Straße, Garching, Germany

⊙ These authors contributed equally to this work.
* sogaard@mpi-marburg.mpg.de

**Data Availability Statement:** All relevant data are within the manuscript and its Supporting Information files.

## Abstract

Cell polarity underlies key processes in all cells, including growth, differentiation and division. In the bacterium *Myxococcus xanthus*, front-rear polarity is crucial for motility. Notably, this polarity can be inverted, independent of the cell-cycle, by chemotactic signaling. However, a precise understanding of the protein network that establishes polarity and allows for its inversion has remained elusive. Here, we use a combination of quantitative experiments and data-driven theory to unravel the complex interplay between the three key components of the *M. xanthus* polarity module. By studying each of these components in isolation and their effects as we systematically reconstruct the system, we deduce the network of effective interactions between the polarity proteins. RomR lies at the root of this network, promoting polar localization of the other components, while polarity arises from interconnected negative and positive feedbacks mediated by the small GTPase MglA and its cognate GAP MglB, respectively. We rationalize this network topology as operating as a spatial toggle switch, providing stable polarity for persistent cell movement whilst remaining responsive to chemotactic signaling and thus capable of polarity inversions. Our results have implications not only for the understanding of polarity and motility in *M. xanthus* but also, more broadly, for dynamic cell polarity.

## Author summary

The asymmetric localization of cellular components (polarity) is at the core of many important cellular functions including growth, division, differentiation and motility. However, important questions still remain regarding the design principles underlying polarity networks and how their activity can be controlled in space and time. We use the rod-shaped bacterium *Myxococcus xanthus* as a model to study polarity and its regulation. Like many bacteria, in *M. xanthus* a well-defined front-rear polarity axis enables efficient translocation. This polarity axis is defined by asymmetric polar localization of a switch-like GTPase and its cognate regulators, and can be reversed in response to signaling cues.

**Funding:** This work was supported by the German Research Council (DFG; https://www.dfg.de/) within the framework of the Transregio 174 "Spatiotemporal dynamics of bacterial cells" (U.G. and L.S.-A.) and the German-Israeli Project Cooperation "Spatial and Temporal Regulation of Macromolecular Complex Formation in Bacteria" (to L.S.-A.) as well as by the Max Planck Society (https://www.mpg.de/de) (to L.S.-A.). The funders had no role in study design, data collection and analysis, decision to publish, or preparation of the manuscript.

**Competing interests:** The authors have declared that no competing interests exist.

Here we use a combination of quantitative experiments and data-driven theory to deduce the network of interactions among the *M. xanthus* polarity proteins and to show how the combination of positive- and negative-feedback interactions give rise to asymmetric polar protein localization. We rationalize this network topology as operating as a spatial toggle switch, providing stable polarity for persistent cell movement whilst remaining responsive to chemotactic signaling and capable of polarity inversions. Our results have broader implications for our understanding of dynamic cell polarity and GTPase regulation in both bacteria and eukaryotic cells.

## Introduction

Most cells display an asymmetric distribution of proteins across cellular space that defines a polarity axis [1]. Cell polarity is key to processes including growth, division, differentiation and motility [1, 2]. Polarity can be stably maintained over time, as in the apical-basolateral polarity of epithelial cells, and stalked *Caulobacter crescentus* cells [3, 4]. Alternatively, polarity can change dynamically in response to external cues, as exemplified by the changing polarity of migrating leukocytes, and front-rear polarity of moving *Myxococcus xanthus* cells [5, 6]. Central questions in cell biology are how local molecular interactions result in the polarized distribution of proteins within a cell and how this polarity can be actively changed over time. Quantitative data analysis together with data driven modelling have recently been harnessed to uncover the principles that underlie the emergence of polarity [7].

In rod-shaped bacteria, the cell poles are key locations for polarized proteins [3]. Three types of cues are known to guide other proteins to the poles: Polar landmark proteins, cell geometry such a negative membrane curvature, and polarly-enriched lipids [2]. Client proteins can remain stably localized at a particular pole during the cell cycle or switch poles dynamically independently of the cell cycle [2, 3].

Rod-shaped *M. xanthus* cells move on surfaces in the direction of their long axis using two motility systems with well-defined front-rear polarity [6, 8]. Type IV pili (T4P)-dependent motility is characterized by cycles of extension, adhesion and retraction of pili at the leading cell pole, thereby enabling forward movement [9]. Gliding motility depends on Agl/Glt complexes that assemble at the leading pole, adhere to the substratum, and propel the cell forward as they relocate towards the lagging pole, where they disassemble [10–18]. With this configuration of motility machineries, cells move persistently in one direction with well-defined leading and lagging cell poles.

Underlying this front-rear polarity axis is a polarity module composed of four proteins: MglA, MglB, RomR and RomX [8, 19–22]. MglA is the key protein of this module, while the other proteins regulate MglA activity (Fig 1A). MglA belongs to the superfamily of small Ras-like GTPases that are central regulators of cell polarity in eukaryotes [5, 23]. These proteins are molecular switches and cycle between an active GTP-bound form, which interacts with downstream effectors to trigger a response [24], and an inactive GDP-bound form. GTPase activity is regulated by Guanine nucleotide Exchange Factors (GEFs) that facilitate the exchange of GDP for GTP, and GTPase Activating Proteins (GAPs) that stimulate the low intrinsic GTPase activity to stimulate formation of the GDP-bound form [25]. Accordingly, MglA-GTP represents the active form of MglA and is essential for both motility systems [21, 22, 26, 27]. MglA-GTP localizes mostly at the leading cell pole while MglA-GDP is delocalized in the cytoplasm [21, 22] (Fig 1A). At the leading pole, MglA-GTP stimulates the T4P machinery by an unknown mechanism and the assembly of Agl/Glt complexes [11]. RomR and RomX form a

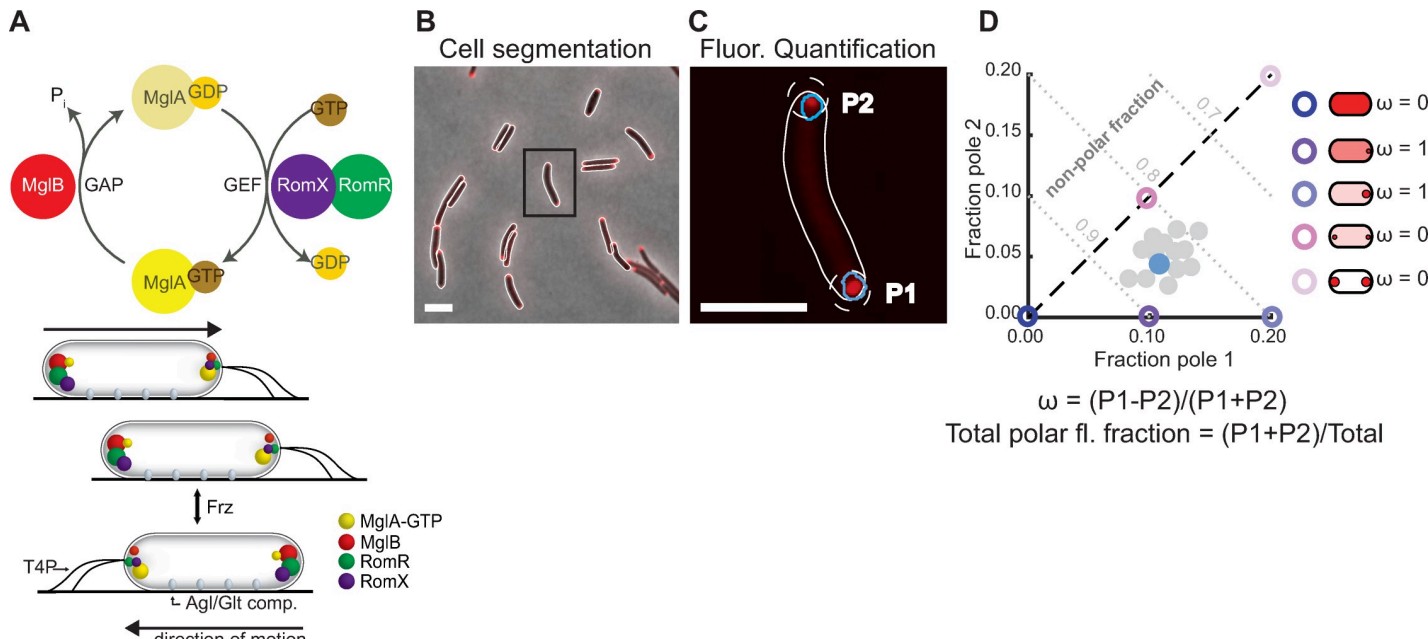

**Fig 1. The polarity module and fluorescence quantification method.** A. MglA GTPase cycle and localization of polarity proteins. T4P are indicated at the leading cell pole and Agl/Glt complexes between cell poles. Amounts of each protein localized at each pole are indicated by the size of the corresponding cluster. B, C. Image quantification pipeline for a representative cell (black rectangle in B). Polar fluorescence clusters (blue outlines) were identified within a search region at each cell pole (white dashed line, Methods). Polar fluorescence was obtained by integrating the fluorescence intensity over each cluster. The pole with higher fluorescence is defined as Pole 1 (P1), the pole with lower fluorescence as Pole 2 (P2). Scale bars, 5 μm. D. Fraction of fluorescence in polar clusters at pole 1 and pole 2 is plotted for individual cells (blue dot: cell in C). Different localization patterns such as symmetric, asymmetrically polarized, or diffuse (right) correspond to distinct regions of polar fraction 1 versus polar fraction 2 space (colored circles). Total polar fluorescence fraction and asymmetry, ω, were calculated as indicated. Note that for cells with P1+P2 = 0, i.e. no detectable polar clusters, ω = 0.

complex that has MglA GEF activity and also acts as a direct MglA-GTP polar recruitment factor [19]. MglB is the cognate MglA GAP [21, 22, 28]. Paradoxically, the GEF RomR/RomX and the GAP MglB localize similarly with a small cluster at the leading and a large cluster at the lagging cell pole (Fig 1A) [19, 21, 22, 29], which would appear to promote a futile cycle of GTP exchange and immediate hydrolysis at the lagging pole.

Importantly, *M. xanthus* cells occasionally reverse their direction of movement. These reversals are induced by the Frz chemosensory system in response to unknown stimuli [30]. The Frz signal to the polarity module is mediated by the response regulators FrzX and FrzZ [31, 32], which localize at the lagging and leading pole, respectively during reversals. During a reversal, MglA-GTP, MglB, RomR and RomX switch poles, resulting in an inversion of the front-rear polarity axis (Fig 1A), and, consequently, the two motility machineries assemble at the new leading pole [12–15, 17, 18, 33, 34].

Molecular-switch proteins are often integral to polarity systems, which can have qualitatively different macroscopic behavior. For example, during bud site selection in *Saccharomyces cerevisiae* by the small GTPase Cdc42, polarity remains static until budding is complete [35], while in cell division site selection by the Min system in *Escherichia coli*, the ATPase MinD oscillates from pole-to-pole [36]. Conceptually, the polarity module of *M. xanthus* can be regarded as a spatial toggle switch: the system can exist in two stable states giving rise to persistent movement with one or other pole being the leading cell pole. At the same time, the system is responsive to signaling by the Frz system that causes inversion of the polarity axis (Fig 1A). The distinct macroscopic behavior of different systems emerges from interactions between individual proteins. Previous work explored how the four proteins of the *M. xanthus* polarity

module interact *in vitro*. MglA-GTP directly interacts with the RomR/RomX complex as well as with MglB [19, 21, 22, 28]. MglB and RomR have been suggested to interact directly [8, 20], but neither RomR nor RomX regulate MglB GAP activity [19]. Because these interactions may be incomplete and, more importantly, *in vitro* analyses of protein-protein interactions rarely suffice to understand the emergent properties of a spatially organized system, it remains an open question how local protein-protein interactions result in the emergence of asymmetric polar localization of the proteins of the polarity module.

The interdependence of polarity protein localization has been analyzed at a general, qualitative level. In the absence of RomR and/or RomX, polar localization of MglA is strongly reduced but not completely abolished [8, 19, 20]. In the absence of RomR and/or RomX, MglB was reported to localize at only one pole. In the absence of MglB, localization of RomR, RomX and MglA was reported to be shifted towards a more symmetric configuration [8, 19–22]. In the absence of MglA, localization of MglB, RomR and RomX are shifted towards localizing at only one pole [8, 19, 20]. Importantly, however, these studies largely relied on qualitative classification of localization patterns, focusing on the presence/absence of polar clusters without separating polar cluster abundance from changes in cluster intensity. This distinction is crucial for fully enumerating the interactions between the polarity proteins *in vivo*.

Here, to uncover the principles underlying the establishment and maintenance of *M. xanthus* cell polarity, we combine quantitative live-cell imaging and data driven theory. By quantifying the localization of the polarity proteins in different combinations, we deduce the interaction network responsible for their polar localization. We find that RomR lies at the root of this network, being principally responsible for polar recruitment of MglA and MglB. The revealed interaction network combines dual positive feedback, as RomR not only stimulates its own polar binding but RomR and MglB mutually recruit each other, and negative feedback, as MglA inhibits RomR/MglB mutual recruitment. Mathematical modeling confirms that these interactions are able to establish the observed polar localization pattern. We rationalize this interaction network as providing stable polarity while remaining responsive to Frz signaling and capable of polarity inversions during cellular reversals.

## Results

To study how the proteins of the polarity module interact *in vivo*, we systematically analyzed their localization dependencies using fluorescently-labelled fusions in live *M. xanthus* cells (Methods). To this end, we developed an image analysis pipeline to precisely quantify polar and cytoplasmic signals of these proteins (Methods). The output of this pipeline is, for each cell, total fluorescence and the fractions of fluorescence in clusters at each pole (Fig 1B, Fig 1C and Fig 1D). Because RomR and RomX form the RomR/RomX GEF complex, Δ*romR* and Δ*romX* mutants have the same phenotype, and RomX displays the same localization pattern as RomR [19], we used RomR localization as a readout for the localization of the RomR/RomX complex, and the effect of a Δ*romR* mutation as a proxy for lack of the RomR/RomX complex. All fluorescent proteins were expressed at wild-type (WT) levels from their native locus unless otherwise noted. While the MglA-mVenus fusion is partially active, the MglB-mCherry and RomR-mCherry fusions are fully active [19, 20].

We first quantified fluorescent protein localization in snapshot images of otherwise WT steady state cultures. Each strain was characterized by determining the mean fraction of fluorescence associated with polar clusters at both poles (mean total polar fluorescence), and the mean asymmetry given by the difference in fluorescence between the two poles normalized by the total polar fluorescence, denoted by ω (Fig 1D). Consistent with prior results, we observed polar localization of all three fluorescent fusion proteins (Fig 2A, Fig 2B and Fig 2C; S1A Fig,

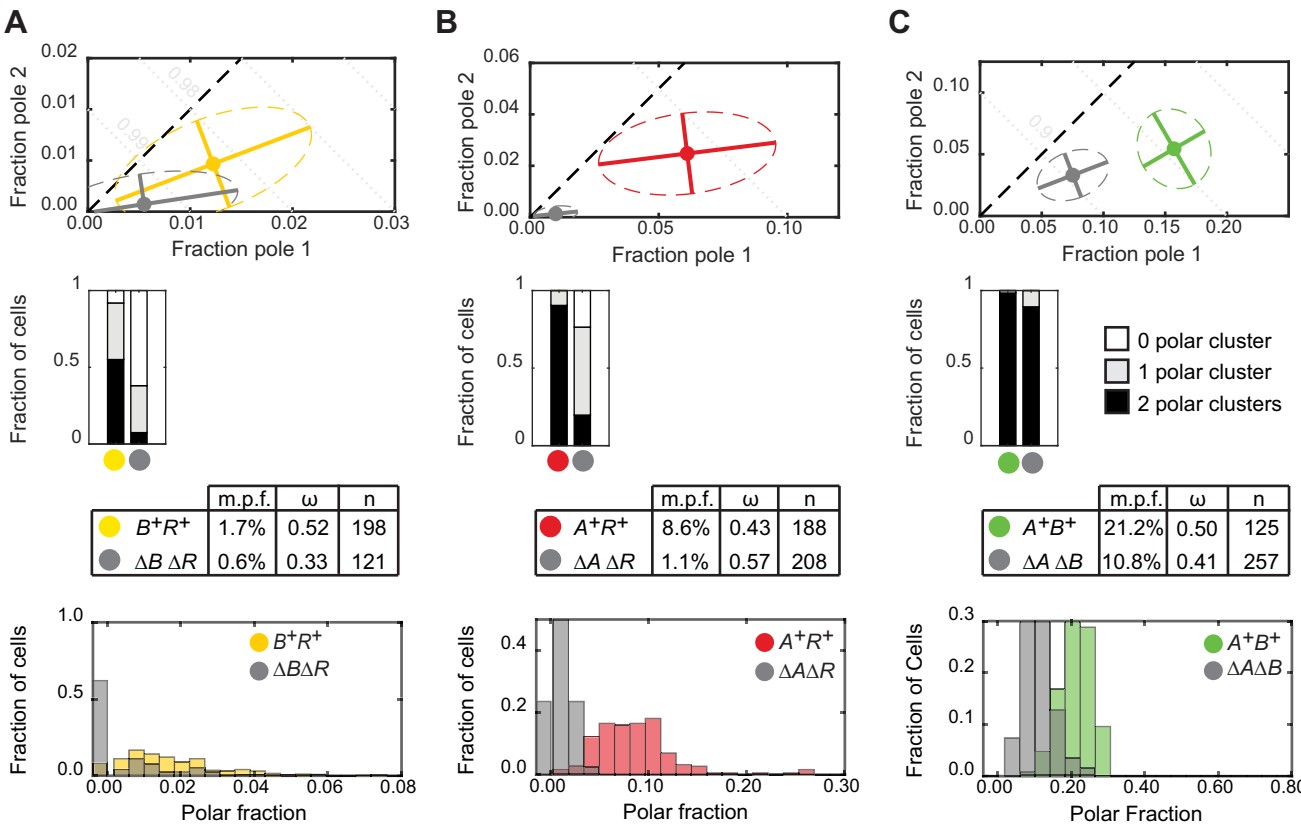

**Fig 2. Quantification of MglA-mVenus, MglB-mCherry and RomR-mCherry polar localization.** A, B and C. Polar localization of MglA-mVenus, MglB-mCherry and RomR-mCherry, respectively, in WT and in the absence of the other two proteins. First row, mean fraction of fluorescence at each pole for cells of indicated strains (filled circles). Dispersion of the single-cell measurements is represented by error bars and ellipses (dashed lines). Direction and length of error bars are defined by the eigenvectors and square root of the corresponding eigenvalues of the polar fraction covariance matrix for each strain. Color code for strains is indicated in row three. Second row, fraction of cells of each strain with two, one or no detectable polar clusters. Third row, mean localization for each strain (m.p.f., mean total polar fluorescence; ω, mean asymmetry; n, number of cells). Fourth row, histograms of the fraction of cells with a given total polar fluorescence. Here and in the following figures, *A*⁺, *B*⁺ and *R*⁺ are short for *mglA*⁺, *mglB*⁺ and *romR*⁺; ΔA, ΔB and ΔR are short for Δ*mglA*, Δ*mglB* and Δ*romR*, respectively.

S1B Fig and S1C Fig; mean total polar fluorescence, MglA-mVenus: 1.7%; MglB-mCherry: 8.6%; RomR-mCherry: 21.2%; mean total polar fluorescence and asymmetry for all strains discussed in this work are summarized in S1 Table). Additionally, all three fluorescent proteins were predominantly asymmetric on average (Fig 2A, Fig 2B and Fig 2C; ω, MglA-mVenus: 0.52; MglB-mCherry: 0.43; RomR-mCherry: 0.50). Of note, in each strain, localization spanned the continuum from unipolar to bipolar symmetric, indicating that polarity protein localization shows a high degree of intrinsic cell-to-cell variability (S1A Fig, S1B Fig and S1C Fig).

## RomR polarizes independently of MglA, MglB and the motility machineries

To determine whether MglA, MglB or RomR individually have the ability to localize at the cell poles, we quantified their localization in the absence of the other two components of the polarity module. For all three proteins, the pattern of polar localization differed significantly from WT in the corresponding double-mutant strain (S1A Fig, S1B Fig and S1C Fig; statistical significance of differences in polar localization distributions are provided in S2 Table). In all cases, the mean total polar fluorescence fraction was significantly reduced (MglA-mVenus:

0.6%; MglB-mCherry: 1.1%; RomR-mCherry: 10.8%; Fig 2A, Fig 2B and Fig 2C; statistical significance of differences in mean total polar fluorescence and mean asymmetry are provided in S3 Table). This reduction was most pronounced for MglB-mCherry, but some polar localization was still observed for all proteins. For RomR-mCherry, the reduction in polar fluorescence was largely due to a reduction in polar cluster intensity (Fig 2C), while for MglA-mVenus and MglB-mCherry we observed both a reduction in polar cluster intensity and in the number of cells with detectable polar clusters (Fig 2A and Fig 2B).

It was reported [8] that MglB-mCherry became more unipolar in the absence of MglA and RomR. We did indeed observe a significant increase in the mean asymmetry in this strain (ω: 0.57). However, this increase resulted largely from the reduced number of polar clusters in this strain (Fig 2B), and therefore the number of cells with clusters at both poles, due to the drastic reduction in total polar fluorescence, highlighting the importance of quantifying polar fluorescence intensity in addition to the qualitative pattern of polar clusters. RomR-mCherry localization was described as symmetric in the absence of MglA and MglB [8]. We observed a significant reduction in the mean RomR-mCherry asymmetry compared to WT (ω: 0.41); however, RomR-mCherry was still asymmetrically localized in most cells (S1C Fig).

Since each of the three proteins still localized polarly to some extent in the absence of the other two, we asked whether this localization could be due to interactions with the polarly localized motility machineries. To this end, we deleted *pilQ* and/or *aglZ*, genes that encode proteins essential for assembly of the T4P machinery and gliding motility machinery, respectively [11, 33], and quantified the resulting localization of MglA, MglB and RomR (S2A Fig, S2B Fig, S2C Fig, S3A Fig, S3B Fig and S3C Fig; statistical significance of differences in polar localization distributions are provided in S2 Table).

Polar MglA-mVenus signals were almost completely eliminated in the Δ*mglB* Δ*romR* Δ*pilQ* Δ*aglZ* mutant (S3A Fig; mean polar fluorescence 0.06%). Compared to Δ*mglB* Δ*romR*, polar fluorescence was also significantly reduced in the Δ*mglB* Δ*romR* Δ*pilQ* mutant (mean total polar fluorescence 0.1%), but the localization pattern was not significantly changed in the Δ*mglB* Δ*romR* Δ*aglZ* mutant (mean total polar fluorescence 1.0%) (S2A Fig and S3A Fig). By contrast, MglA-mVenus was still strongly polar in the presence of MglB and RomR in a Δ*pilQ* strain, although the mean polar fluorescence was slightly but significantly reduced (1.3%) compared to WT (S2A Fig and S3A Fig). We conclude that the polar T4P machinery is sufficient for polar localization of MglA in the absence of MglB and RomR, but plays only a marginal role in the presence of MglB and RomR. For MglB-mCherry, the mean polar fluorescence in the Δ*mglA* Δ*romR* Δ*aglZ* Δ*pilQ* mutant (1.2%) was not significantly different from that in the Δ*mglA* Δ*romR* mutant (S2B Fig and S3B Fig) supporting that the small, residual polar localization of MglB-mCherry in the absence of MglA and RomR is not due to either of the motility machineries. RomR-mCherry mean total polar fluorescence was also not significantly different in the Δ*mglA* Δ*mglB* Δ*aglZ* Δ*pilQ* mutant (mean polar fluorescence 11.5%) from the Δ*mglA* Δ*mglB* mutant (S2C Fig and S3C Fig).

Comparing the localization patterns of the proteins of the polarity module in WT to those observed in the absence of the other two components and the motility machineries, we conclude that only RomR-mCherry has the ability to significantly localize at the poles in isolation, suggesting that RomR is at the root of the interactions that result in polar localization of MglA and MglB. While polar RomR was predominantly asymmetric in the absence of MglA and MglB, some amount was nevertheless present at both poles in almost all cells (S3C Fig), thereby potentially providing the capacity to recruit both MglA and MglB at opposite poles.

## RomR polarizes stably and asymmetrically, independently of MglA and MglB

Given that RomR likely plays a critical role in polar localization of MglA and MglB, and since polar RomR-mCherry largely remained asymmetric in the absence of both MglA and MglB in snapshot images, we next asked whether RomR asymmetry is stably maintained at the time-scale of the cell cycle (~6 hrs), or whether it was dynamic on shorter timescales. To this end, we performed time-lapse recordings of WT and Δ*mglA* Δ*mglB* cells containing RomR-mCherry with images recorded every 10 min for 6 hrs. While WT cells showed frequent and rapid polarity inversions (39%, defined as events when the cell pole with a weaker fluorescent signal in one frame became the pole with the stronger signal in the following frame), Δ*mglA* Δ*mglB* cells did not show such clear inversion events and had a significantly lower polarity inversion probability (12%; $\chi^2$ test for independence, $p \ll 10^{-10}$) (Fig 3A; S4A Fig and S4B Fig; S4 Table). Autocorrelation functions for the polar fractions confirmed that changes in polar fluorescence were significantly slower in the Δ*mglA* Δ*mglB* mutant than in WT (S4C Fig). Thus, asymmetric polar RomR localization is established and stably maintained (relative to WT) in the absence of MglA and MglB.

We next sought to understand whether this RomR asymmetry arose spontaneously or whether it reflected an underlying asymmetry between the poles. Since *M. xanthus* divides by binary fission, giving rise to daughter cells each with an old and a new cell pole, an obvious candidate for asymmetry is new- versus old-pole identity. We identified cell division events during the time-lapse recordings and quantified RomR-mCherry localization in newborn cells. In the absence of MglA and MglB, RomR-mCherry localization correlated significantly with the old pole (64% of cells had the largest RomR-mCherry cluster at the old pole immediately after division) and that bias persisted for several hrs (Fig 3B; S4A Fig). By contrast, in WT we observed a weak preference for the new pole immediately after cell division (61% of cells had the largest RomR-mCherry cluster at the new pole) but this bias was rapidly lost due to the frequent and asynchronous switching events (Fig 3B; S4B Fig). We conclude that RomR polar localization correlates with the old pole in the absence of MglA and MglB, but this inherent asymmetry is not observed in the WT.

## RomR accumulates cooperatively at the poles

The experiments discussed above document that under steady state conditions and in the absence of MglA and MglB, RomR localizes stably and largely asymmetrically at the cell poles, with a preference for the old pole. Next, we asked how polar localization of RomR in the absence of MglA and MglB is established. To this end, we constructed a Δ*mglA* Δ*mglB* Δ*romR* strain in which *romR-mCherry* was expressed from a vanillate-inducible promoter, and investigated by time-lapse microscopy RomR-mCherry localization upon induction. Cells were grown in suspension in the absence of vanillate and then placed on an agar surface containing 300 μM vanillate and imaged every 15 min for 6 hrs. RomR-mCherry was undetectable in immunoblots in the absence of vanillate, but accumulated gradually in the presence of 300 μM vanillate, reaching a level slightly higher than when expressed from the native promoter after 6 hrs (S5A Fig). Total cell fluorescence increased over time in agreement with the immunoblots (S5A Fig). Consequently, we used an estimate of the RomR-mCherry concentration during induction in which total cellular fluorescence was normalized by the cell area calculated from phase contrast images, which was used as a proxy for cell volume. We refer to this metric as "fluorescence concentration". Since the exchange timescale of RomR in polar clusters determined by fluorescence-recovery-after-photobleaching (FRAP) is significantly shorter (~28s)

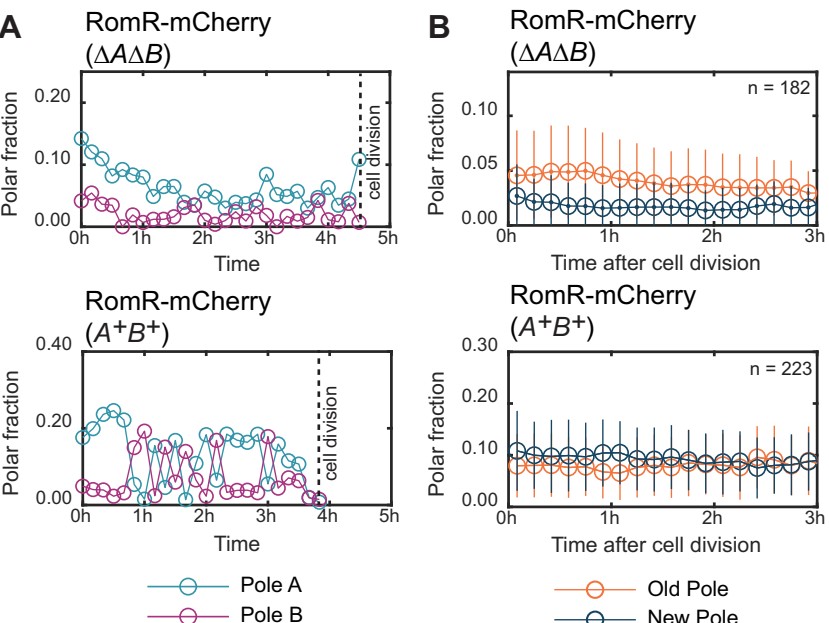

**Fig 3. RomR polar localization is stably maintained and correlates with the old pole in the absence of MglA and MglB.** A. Dynamics of RomR-mCherry polar fluorescence in representative single cells of the Δ*mglA* Δ*mglB* Δ*aglQ* (top) and Δ*aglQ* (bottom) strains. The Δ*aglQ* mutation was introduced to inactivate the gliding motility machinery and enable recordings of the same cells for several hrs. Pole A and B are defined as the pole with the highest and lowest fluorescence in the first frame, respectively. B. Fraction of RomR-mCherry fluorescence at the old (orange) and new (blue) cell pole as a function of time after cell division in the Δ*mglA* Δ*mglB* Δ*aglQ* (top) and Δ*aglQ* (bottom) strains. Plotted are the mean ± one standard deviation of all observed cells at each time point. n: number of cells observed immediately after division. Because cells divide at different time points during the recording period, the number of cells included at each time point varies; however, at least 40 cells were included per time point. In the absence and presence of MglA/MglB, RomR-mCherry localization was significantly biased (two-sided binomial test, p = $10^{-4}$ and p = 0.002, respectively) towards the old pole and new pole, respectively. The asymmetries in the two strains also differed significantly ($\chi^2$ test for independence, p = $10^{-6}$).

[31]) than our imaging interval, we assume that the observed polar RomR-mCherry localization reflects steady-state localization at the corresponding concentration.

We observed asymmetric polar accumulation of RomR-mCherry in the Δ*mglA* Δ*mglB* Δ*romR* strain at all fluorescence concentrations (Fig 4A). Furthermore, once polarity was established in an individual cell, its direction remained largely stable (S5D Fig). From 75 min after the start of induction, the first time point where most cells had at least one polar cluster, until the end of the experiment, we observed a polarity inversion probability of 14%, similar to that observed for RomR-mCherry expressed from the native promoter in Δ*mglA* Δ*mglB* cells ($\chi^2$ test for independence, p = 0.08).

Importantly, the shape of the polar accumulation curves (Fig 4A) provides evidence for positive cooperativity in RomR-mCherry polar localization: In the absence of any cooperativity, individual RomR-mCherry molecules would localize independently of one another, such that the fraction at each pole should be constant and independent of concentration. Instead, the fractions of RomR-mCherry at both poles increased with fluorescence concentration, suggesting that RomR-mCherry self-recruits or stabilizes its polar accumulation.

We also asked whether RomR-mCherry polar localization during induction showed a similar old-pole bias as in the steady state experiments. Therefore, we repeated the *romR-mCherry* induction experiment in a Δ*mglA* Δ*mglB* Δ*romR* strain co-expressing PilQ-sfGFP, which localizes stably with the largest cluster at the old pole in 91% of cells after cell division in this genetic

**A**

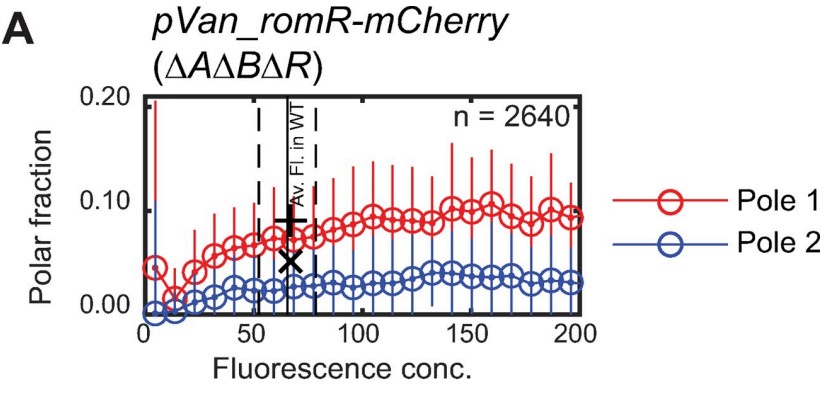

*pVan_romR-mCherry*
(ΔAΔBΔR)

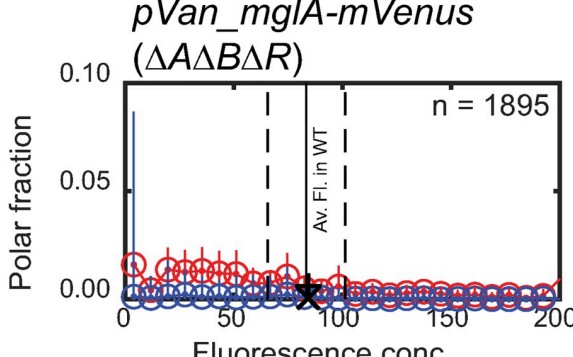

*pVan_mglA-mVenus*
(ΔAΔBΔR)

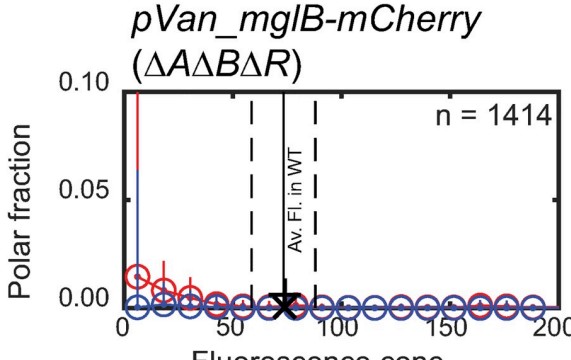

*pVan_mglB-mCherry*
(ΔAΔBΔR)

**B**

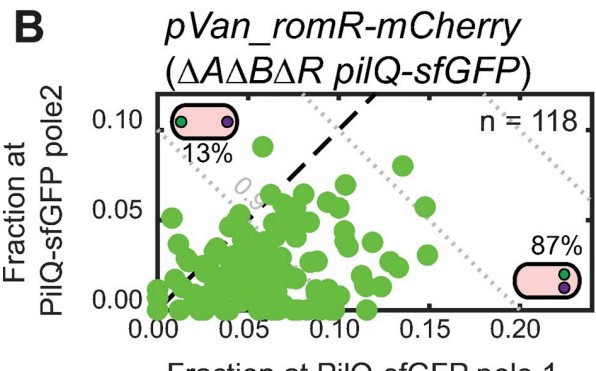

*pVan_romR-mCherry*
(ΔAΔBΔR *pilQ-sfGFP*)

**Fig 4. RomR accumulates cooperatively at the cell poles.** A. Induction of *romR-mCherry*, *mglA-mVenus* and *mglB-mCherry* expression from the vanillate inducible promoter in strains of the indicated genotypes. Δ*aglQ* Δ*frzE* mutations were introduced in all strains to allow monitoring of the same cells for several hours and to reduce Frz-dependent polarity inversions. n, number of individual cell observations. Cells from all frames of the time-lapse recordings were pooled and binned according to their fluorescence concentration (total cellular fluorescence divided by cell area). Plotted are the mean ± one standard deviation of all cells within each bin. Each bin contain data from at least five cells. Vertical lines indicate mean ± one standard deviation of the fluorescence concentration of the same protein expressed from the native site in the Δ*aglQ* Δ*frzE* background, and imaged under the same conditions. Polar fractions calculated from snapshots of this strain are marked (+: pole 1, x: pole 2). B. RomR-mCherry preferentially accumulates at the old pole during induction. *romR-mCherry* was induced in a Δ*mglA* Δ*mglB* Δ*romR* Δ*aglQ* Δ*frzE* *pilQ-sfGFP* strain. After 2 hrs, pole identities were assigned according to polar PilQ-sfGFP signals (i.e. pole 1 is the pole with greater PilQ-sfGFP signal). RomR-mCherry localization was plotted using the pole 1 and 2 identities determined based on PilQ-sfGFP (green dots). Cells in which the higher RomR-mCherry and PilQ-sfGFP fluorescence coincided lie below the dashed line (see inset representations, green: RomR-mCherry; purple: PilQ-sfGFP); cells in which higher RomR-mCherry and PilQ-sGFP fluorescence occurred at opposite poles lie above the dashed line. RomR-mCherry/PilQ-sfGFP colocalization was significant (one-sided binomial test, $p \ll 10^{-10}$).

background (S6 Fig). After 2 hrs of induction, in 87% of cells that had not yet undergone cell division the brighter RomR-mCherry and PilQ-sfGFP clusters coincided at the same pole (Fig 4B) indicating that *de novo* synthesized RomR-mCherry clusters form preferentially at the old pole. This suggests that the old-pole bias in RomR-mCherry localization in the Δ*mglA* Δ*mglB* background is not due to pre-existing RomR asymmetry inherited from the mother cell, but instead reflects an intrinsic preference for RomR recruitment at this pole. However, since this preference appears to be overcome in WT, likely by the interactions with MglA and/or MglB, we did not investigate further the mechanism underlying this old pole preference.

As expected, in similar experiments in which MglA-mVenus or MglB-mCherry were induced from the vanillate inducible promoter in the relevant double mutants, we observed only weak polar protein localization at all fluorescence concentrations and found no evidence of cooperative polar accumulation (Fig 4A; S5B Fig and S5C Fig).

## Rebuilding the polarity module

Having determined how each component of the polarity module behaves in isolation, we next investigated the interactions between them by studying how polar localization patterns changed as the polarity system was systematically reassembled from its individual components. To this end, we analyzed snapshots of steady-state cells natively expressing MglA-mVenus, MglB-mCherry, or RomR-mCherry in the relevant double- and single-deletion backgrounds as well as in WT.

Starting from the Δ*mglB* Δ*romR* mutant, detectable MglA-mVenus polar fluorescence was significantly reduced upon addition of MglB (mean total polar fluorescence 0.02%; Fig 5A; S1A Fig), while polar localization increased dramatically with the addition of only RomR (mean total polar fluorescence 8.3%). WT localization (mean total polar fluorescence 1.7%) was intermediate between that observed in the presence of RomR or MglB individually. These observations are consistent with RomR/RomX being a GEF and recruiting MglA-GTP to the poles, and MglB inhibiting MglA polar recruitment by converting MglA-GTP to MglA-GDP (Fig 5A).

Starting from the Δ*mglA* Δ*romR* mutant, MglB-mCherry mean polar fluorescence decreased marginally but significantly with the addition of MglA (0.5%), but polar MglB-mCherry increased dramatically upon addition of RomR (mean polar fluorescence: 20.3%) (Fig 5B; S1B Fig). WT localization was intermediate between these two conditions (mean polar fluorescence: 8.6%). Polar localization was also significantly more asymmetric in the presence of RomR only (ω: 0.64) than in WT (0.43) (Fig 5B; S1B Fig). These observations suggest that

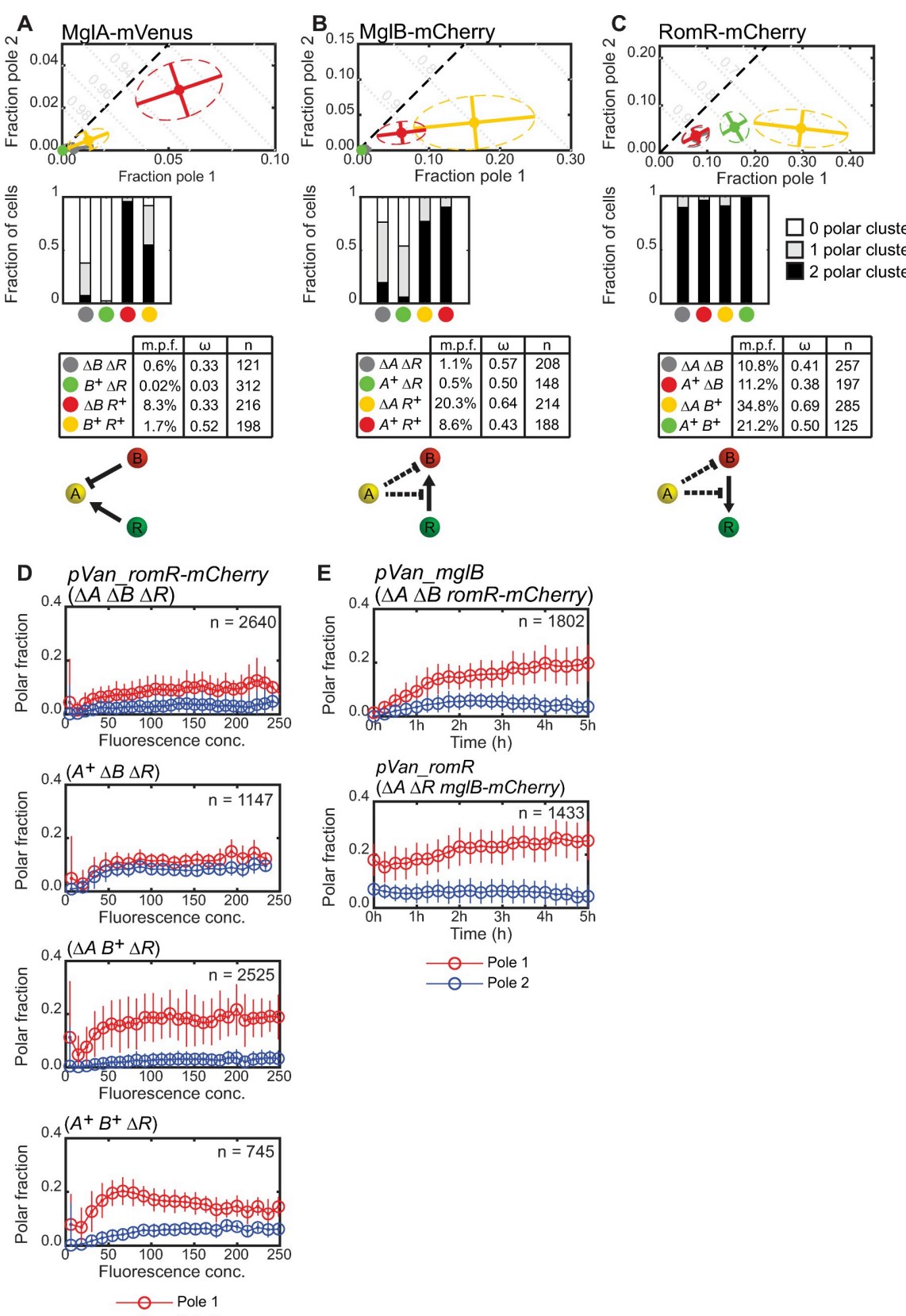

**Fig 5. Rebuilding the polarity module.** A, B and C. Polar localization of MglA-mVenus, MglB-mCherry and RomR-mCherry in WT and in the absence of one or both of the other proteins. Data are presented as in Fig 2A–2C. Note that the data for the WT and double mutants are the same as in Fig 2A–2C. Fourth row, interactions inferred from the changes in polar localization. Positive interactions are represented by pointed arrows, and negative interactions by blunt arrows. Dashed lines indicate possible alternative interactions that cannot be distinguished based on the available data. D. Induction of *romR-mCherry* in strains of the indicated genotypes. All strains also contained Δ*aglQ* Δ*frzE* mutations as in Fig 4A. Polar fluorescence fractions are plotted against fluorescence concentration, as described for Fig 4A. Data in the upper panel are the same as in Fig 4A. E. Induction of *mglB* or *romR*. Top, RomR-mCherry localization is plotted over time upon induction of *pVan_mglB* in the Δ*mglA* Δ*mglB romR-mCherry* strain. Bottom, MglB-mCherry localization is plotted over time upon induction of *pVan_romR* in the Δ*mglA* Δ*romR mglB-mCherry* strain. Both strains also contained Δ*aglQ* Δ*frzE* mutations as in Fig 4A.

RomR enhances, and MglA inhibits MglB polar localization, although the latter effect is most clearly evident in the presence of RomR (Fig 5B).

Starting from the Δ*mglA* Δ*mglB* mutant, RomR-mCherry localization did not change significantly in the presence of MglA (mean polar fluorescence: 11.2%, ω: 0.38), but polar RomR-mCherry increased dramatically and became highly asymmetric in the presence of MglB only (mean polar fluorescence: 34.8%, ω: 0.69) (Fig 5C). Once again, WT localization was intermediate between the two single mutants (mean polar fluorescence: 21.2%, ω: 0.50). These data show that MglB helps to recruit RomR, while MglA tends to disperse RomR but only in the presence of MglB (Fig 5C).

These observations are largely consistent with previous studies of polarity protein localization [8, 19, 20], although there are some discrepancies. Both MglA and RomR were previously described as largely symmetrical in the absence of MglB. We observed a slight but significant reduction in mean asymmetry for MglA-mVenus in Δ*mglB* (ω: 0.33) compared to WT (ω: 0.52), however, most cells remained asymmetric. Likewise RomR asymmetry in Δ*mglB* (ω: 0.38) was slightly but significantly reduced in comparison to WT (ω: 0.50), however, most Δ*mglB* cells still had asymmetric RomR-mCherry localization. It was also reported that MglB localized to one pole in the absence of RomR. However, as for the Δ*mglA* Δ*romR* strain discussed previously, we conclude that this is a consequence of the dramatic reduction in polar MglB-mCherry in the Δ*romR* strain, which dramatically reduces the number of cells with detectable clusters at both poles.

Importantly, previous analyses did not quantify changes in polar cluster intensity, such as the dramatic increase in polar MglB-mCherry and RomR-mCherry in the absence of MglA. These observations in particular suggest a positive feedback between MglB and RomR for mutual polar accumulation, in addition to the positive feedback of RomR on itself. To further test the idea of MglB and RomR mutual recruitment, we repeated the RomR-mCherry induction experiment in the presence of MglB and/or MglA. In the presence of MglA, RomR-mCherry accumulated at the poles to similar levels as in the Δ*mglA* Δ*mglB* background, albeit more symmetrically (Fig 5D). By contrast, in the presence of MglB only, RomR-mCherry accumulated more asymmetrically with the brighter pole accounting for a larger fraction of fluorescence. Finally, in the presence of both MglA and MglB, there was decreased asymmetry between the polar fractions compared to the MglB-only strain. Consistently, we also observed that upon induction of MglB in the Δ*mglA* Δ*mglB* strain, polar accumulation of natively-expressed RomR-mCherry at one pole increased and cells became more asymmetric (Fig 5E, upper panel); and, upon induction of RomR in a Δ*mglA* Δ*romR* strain, natively-expressed MglB-mCherry became strongly asymmetrically polar (Fig 5E, lower panel). These observations are consistent with the interactions deduced from the steady state measurements and support that RomR and MglB mutually recruit each other.

## A mathematical model reproduces steady-state polar localization

Based on our experimental observations and previously published data, we conclude that (1) RomR accumulates at the poles asymmetrically and cooperatively, even in the absence of the MglA and MglB. (2) MglA is polar only in the GTP-bound state and the RomR/RomX complex promotes polar localization of MglA in WT, partly through its GEF activity and partly as a polar recruitment factor. (3) RomR is central to MglB polar localization; RomR and MglB stimulate polar recruitment of one another, thereby establishing a positive feedback in localization. (4) MglB reduces MglA-GTP polar binding through its GAP activity. (5) In the presence of both MglB and RomR, MglA dramatically decreases polar localization of both proteins, suggesting that MglA disrupts the positive feedback between MglB and RomR. However, our data cannot identify the mechanism of this disruption. It may be that MglA suppresses the interactions between MglB and RomR (Fig 6A, blunt blue arrow). Alternatively, MglA may directly reduce polar accumulation of MglB, and thereby indirectly affect accumulation of RomR Fig 6A, blunt orange arrow. To test these possibilities, and to determine whether the major interactions outlined above (Fig 6A) are sufficient to give rise to the emergent asymmetry of the polarity proteins, we turned to quantitative modeling.

Recently, Guzzo et al. [31] introduced a mathematical model of the MglA-MglB-RomR system in *M. xanthus* that reproduced WT localization of the polarity proteins using a proposed interaction scheme deduced from previously reported localization patterns. However, as discussed above, these localization patterns differed in several key ways from our observations. Most notable is that MglB was previously reported to be highly asymmetric in the absence of RomR, or of RomR and MglA, from which Guzzo et al. inferred that MglB should cooperatively self-assemble at one pole. Our data indicates instead that MglB polar localization is greatly reduced in the absence of RomR, and that RomR and MglB recruit one another to the poles. Therefore, we adapted the model of Guzzo et al. to incorporate the interactions that we have documented above (Fig 6A and Fig 6B; S5 Table; Methods). Notably, our data suggest that RomR polar asymmetry in mutant strains, but not WT, reflects an underlying preference for the old cell pole (Fig 3; S4 Fig and S7 Fig). We modeled this by directly implementing a bias in the polar binding affinities of RomR in all mutant conditions but not in WT.

We used this model to test different potential modes of action for MglA in parameter regimes consistent with the experimental steady-state localization patterns in WT and single- and double-mutant strains (see Methods). We found that in a model where MglA acted to suppress recruitment of RomR by MglB, the correct WT polarity pattern was stably established (Fig 6C, blue; S1 Text). By contrast, when MglA directly enhanced the dissociation of MglB, WT polarity could not be established or maintained and the system always evolved to a symmetric state (Fig 6C, orange; S1 Text). Combining the two effects of MglA, provided only a slight improvement in the agreement between model and experiment in the polar localization patterns across the set of mutant strains compared to MglA acting to suppress recruitment of RomR by MglB only. We conclude that the principal mode of action of MglA is to suppress the mutual recruitment of RomR and MglB, while direct regulation on MglB plays only a minor role.

The quantitative agreement between the model and experimental data (Fig 5A, Fig 5B and Fig 5C vs Fig 6D) indicates that the proposed interactions deduced from our *in vivo* analyses (Fig 6A) are sufficient to explain the observed polarity patterns. While our model does not rely on and cannot account for precise molecular interaction mechanisms, it nevertheless elucidates the principles behind asymmetric polar localization (Fig 6E). An initial asymmetry in the polar abundance of any of the proteins is amplified by the combination of positive feedbacks in RomR/MglB recruitment and negative feedback from MglA to disrupt the RomR/MglB

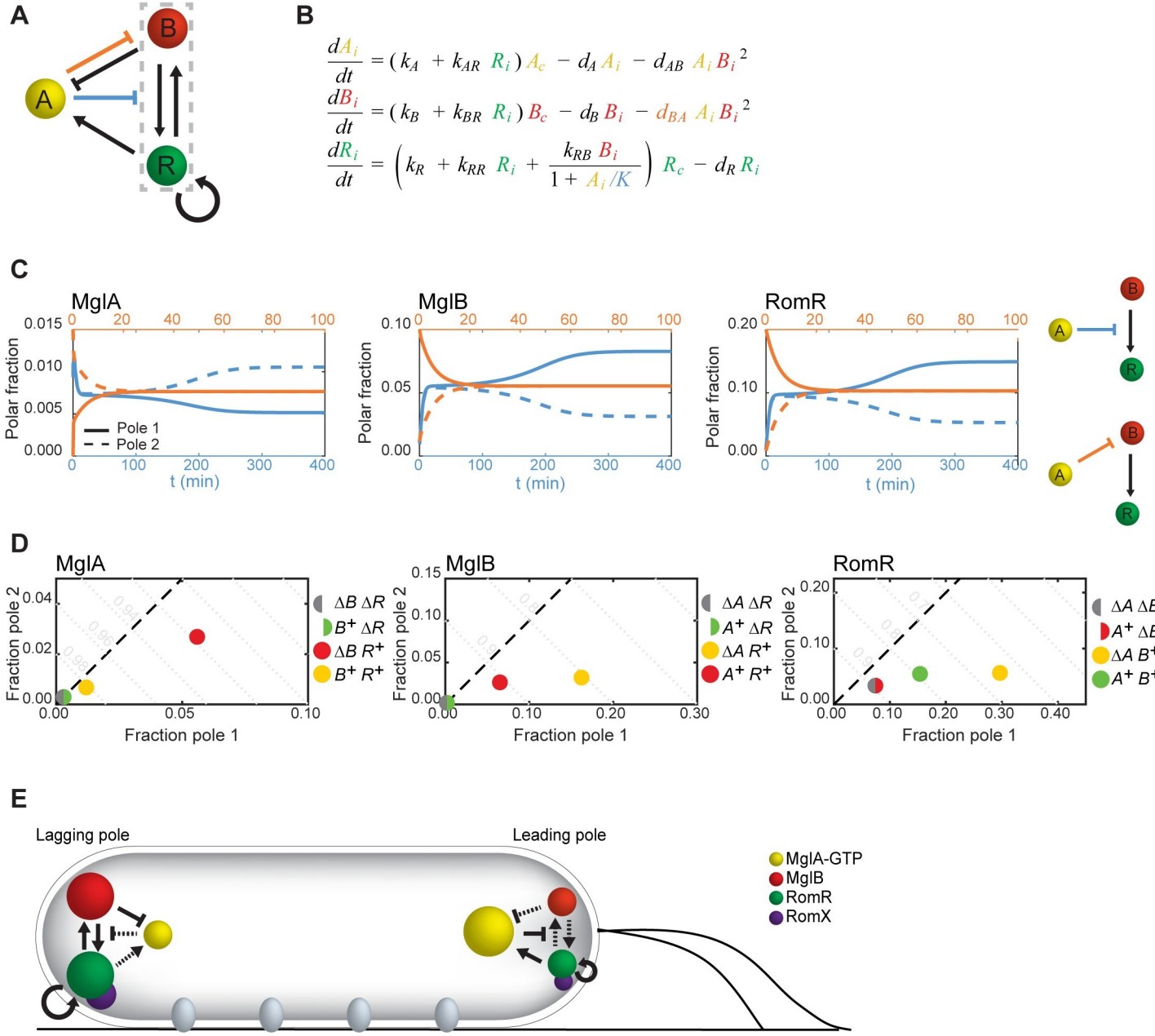

**Fig 6. Mathematical model of the polarity module.** A. Summary of the interactions inferred from experiments in Figs 4 and 5. Dashed box highlights the positive feedback between MglB and RomR. Blue and orange indicate possible modes of actions of MglA on polar localization of MglB and RomR. B. Equations of the mathematical model. Model variables $X_i$ represent the fraction of species $X$ localized at pole $i = 1$ or 2, and $X_c = 1-X_1-X_2$ is the non-polar fraction. $A_i$, $B_i$, $R_i$ represent the fraction of MglA, MglB and RomR/RomX, respectively, localized at pole $i$ (where $i = 1$ or 2). The corresponding non-polar fractions are represented by $A_c$, $B_c$, and $R_c$, where $X_c = 1+X_1-X_2$. C. Dynamics of mathematical model with different modes of action for MglA (indicated by colors in the corresponding network diagrams to the right). Solid and dashed lines indicate the polar fractions of each protein at the two poles. Pole 1 and pole 2 are defined by the localization of MglB and RomR. When MglA suppresses recruitment of RomR by MglB, polarity is established from a small initial asymmetry ($X_1(0) = 0.011$, $X_2(0) = 0.01$ for $X = A,B,R$). When MglA enhances MglB dissociation, asymmetry is lost and the cell becomes symmetric ($X_i(0)$ set to the WT mean polar fractions in snapshot experiments, Figs 2A and 5A, but with MglA polarity inverted relative to MglB and RomR). Parameter values are given in S5 Table. D. Steady state polar fractions produced by the combined mathematical model with both modes of action for MglA, in WT, single- and double-mutant conditions. E. Different interactions dominate at the leading and lagging poles. Full arrows show locally strong interactions, dashed arrows show interactions that are locally suppressed. Amounts of each protein localized at each pole are indicated by the size of the corresponding cluster.

positive feedback. In this way, an excess of MglB and RomR at one pole will grow while displacing MglA. MglA can become stably established at the opposite pole with the help of the small amount of RomR that will intrinsically self-assemble there, and, in turn, limits the accumulation of RomR/MglB at this pole.

## RomR determines dynamic polarity establishment

Finally, we investigated how the future polarity direction was determined during the establishment of polarity. To this end, we studied the dynamics of the model when initialized with a preexisting asymmetry in two of the proteins simultaneously (Fig 7A, S8 Fig).

When a simulated cell was initialized in one of the three possible configurations consistent with WT polarity (i.e. either MglB or RomR at one pole and MglA at the other, or MglB and RomR co-localized at one pole and no protein at the opposite pole), the system evolved straightforwardly to the expected final configuration (the former pole becoming the new lagging pole and the latter pole becoming the new leading pole, S8 Fig). However, it was less clear how pole identities would develop when the system was initialized in a configuration that is not consistent with the WT arrangement. When MglA and either MglB or RomR were initially colocalized, this pole became the new lagging pole, as in the case of MglB or RomR asymmetry alone (Fig 7A). Thus, in both cases, RomR-MglB positive feedback became established at this pole, overcoming inhibition by MglA. Interestingly, when RomR and MglB were initially located at opposite poles, we observed that the pole where RomR was present became the future lagging pole (Fig 7A). Importantly, a small RomR asymmetry can also overcome a larger initial MglB asymmetry (e.g. with 1% of MglB at one pole, 0.2% of RomR at the opposite pole is sufficient to define the latter as the future lagging pole). These findings again identify RomR as the core of the polarity network, and our model predicts that initial RomR accumulation is the dominant factor in determining the future polarity direction, with the pole at which RomR initially accumulates becoming the new lagging pole. To test this prediction, we conducted induction experiments in motile cells and used direction at the onset of movement to identify the newly established leading and lagging poles.

We first considered *mglA* induction. At the start of induction, large MglB-mCherry and RomR-mCherry clusters were observed at one pole (Fig 7B). Our model predicts that the pole at which MglB and RomR are already present will become the new lagging pole, and MglA will accumulate at the (opposite) new leading pole. As expected, at the onset of movement MglB-mCherry (100% of tracked cells) and RomR-mCherry (95% of tracked cells) clusters were significantly biased towards the newly established lagging pole (Fig 7B and Fig 7C). In separate experiments where MglA-mVenus was induced, accumulation of this protein was significantly biased towards the new leading pole (78% of tracked cells) (Fig 7B and Fig 7C), confirming our prediction.

In the case of RomR induction, polar levels of MglA and MglB are initially much lower (Fig 7D). Our model predicts that the initial polar accumulation of RomR will define the new lagging pole, and will dominate over any preexisting MglA or MglB asymmetry. Indeed, upon induction of *romR-mCherry*, we observed a significant bias towards the lagging pole, with 96% of cells having higher levels of RomR-mCherry fluorescence at the new lagging pole at the onset of movement (Fig 7D and Fig 7E). In separate *romR* induction experiments, the larger MglA-mVenus clusters were biased towards the newly established leading pole at the onset of movement (75% of cells); importantly, at the onset of movement, we observed no significant bias in the location of the larger MglB-mCherry cluster (Fig 7D and Fig 7E). This lacking bias was transient, as MglB-mCherry subsequently relocated to the lagging pole in all cells. These results support our predictions that RomR asymmetry is key to establishing the new polarity

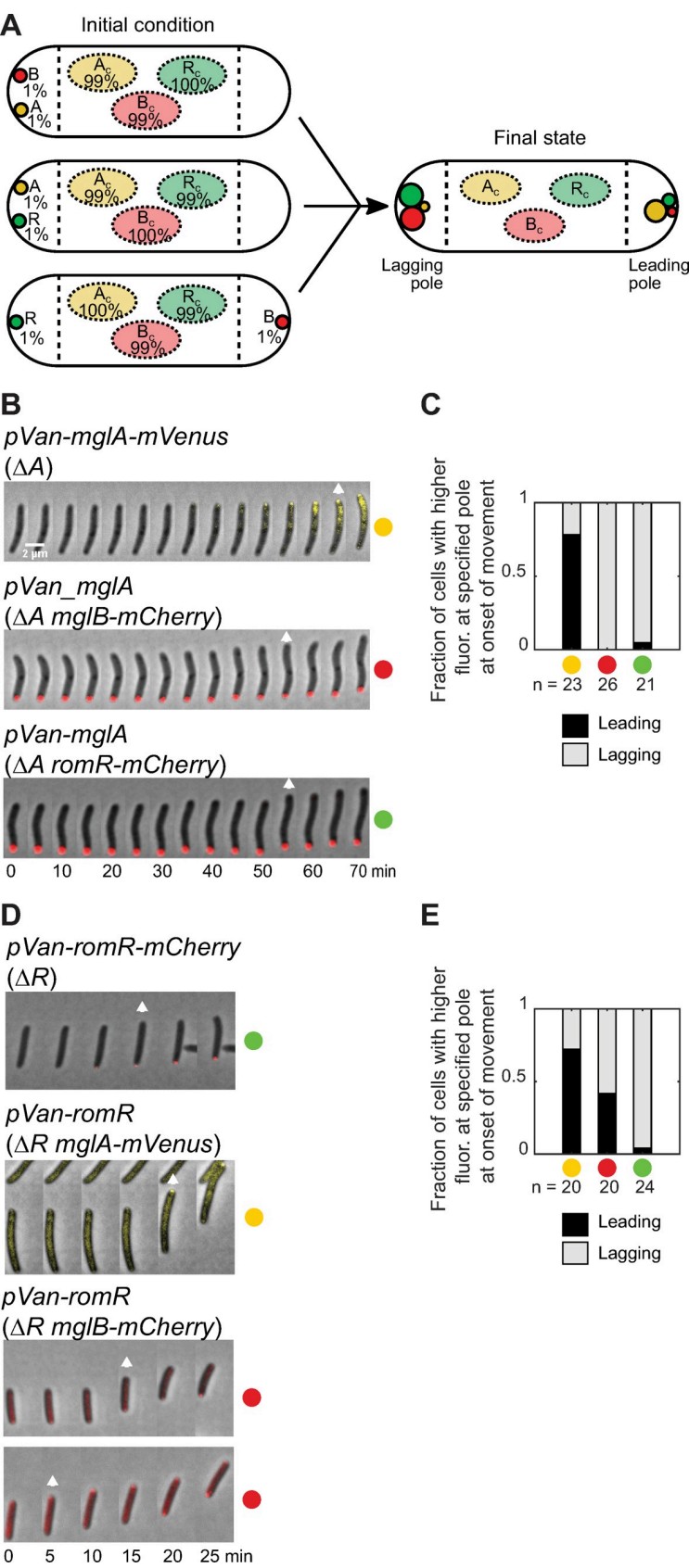

**Fig 7. Exploring the dynamic establishment of polarity at the onset of movement.** A. Polarity establishment based on the mathematical model. Simulated cells were initialized with polar asymmetry (1%) of two proteins, as indicated (left). For each of the initial arrangements shown, the system evolves to the same final state (right). In particular, if RomR and MglB are initially at opposite poles, the pole with RomR becomes the future lagging pole. B. Localization of MglB-mCherry (top), RomR-mCherry (middle) and MglA-mVenus (bottom) at the onset of movement during induction of *mglA* (top, middle) or *mglA-mVenus* (bottom) with 300 μM vanillate for the indicated period of time. White arrow indicates onset of movement. Scale bar, 2 μm. C. Fraction of cells in B in which the brighter MglA-mVenus (yellow), MglB-mCherry (red) and RomR-mCherry (green) polar clusters were at the indicated pole at the onset of movement. MglA-mVenus, MglB-mCherry and RomR-mCherry were all significantly biased (two-sided binomial tests, p = 0.011, p = $3×10^{-8}$ and p = $2×10^{-5}$) towards the leading, lagging and lagging poles, respectively. n, number of cells analyzed. D. Localization of RomR-mCherry (top), MgA-mVenus (middle) and MglB-mCherry (bottom) at the onset of movement during induction of *romR-mCherry* (top) or *romR* (middle, bottom) with 300 μM vanillate for the indicated period of time. White arrow indicates onset of movement. E. Fraction of cells in D in which the brighter MglA-mVenus (yellow), MglB-mCherry (red) and RomR-mCherry (green) polar clusters were at the indicated pole at the onset of movement. MglA-mVenus and RomR-mCherry were significantly biased (two-sided binomial tests, p = 0.041 and p = $3×10^{-6}$) towards the leading and lagging pole, respectively, while MglB-mCherry did not show a leading/lagging pole bias (two-sided binomial test, p = 0.82). n, number of cells analyzed.

direction, and that this direction is chosen largely independently of existing MglA and MglB asymmetry.

## Discussion

Here, we uncover the principles underpinning front-rear polarity in *M. xanthus*. To understand the contribution of each component of the polarity module, we untangled the system and examined each component in isolation, using precise *in vivo* techniques to quantify subcellular localization, combined with *in silico* methods. Our approach revealed the topology of (direct or indirect) interactions (Fig 6A) that allow MglA, MglB and RomR to localize asymmetrically at the poles.

Our data provide evidence that RomR is the key protein responsible for polar recruitment of MglA and MglB. RomR is always polar, independently of the presence of MglA, MglB, or the motility machineries. Moreover, RomR is still significantly asymmetric in isolation, and induction experiments revealed that RomR alone accumulates cooperatively at the poles. MglA localizes to the poles due to the GEF activity of the RomR/RomX complex and direct recruitment of MglA-GTP by polar RomR/RomX [19]. No evidence supports that MglA stimulates RomR binding, thus, excluding a RomR-MglA positive feedback. RomR enhances MglB polar accumulation and *vice versa;* thus, polar accumulation of RomR and MglB positively feedback on one another. MglB, as the MglA-GAP, also reduces MglA polar accumulation by stimulating GTP hydrolysis by MglA. Finally, we observed that MglA also decreases RomR and MglB polar accumulation in the presence of both proteins. While the exact molecular mechanism is not understood, we speculate that MglA might interfere with the interaction between MglB and RomR and thereby disrupt the positive feedback in MglB-RomR mutual recruitment. Consistent with its role as the primary polar localization factor, our mathematical model suggests that establishment of polarity is highly sensitive to asymmetry in RomR accumulation, which can overcome a preexisting asymmetry in MglA or MglB to determine the polarity direction. This is supported by our induction experiments, where we observed that RomR accumulation defines the new lagging pole, largely independently of the existing localization of MglB and/or MglA.

To understand how these interactions give rise to emergent cell polarity, we asked about the origin of symmetry breaking in the *M. xanthus* polarity module. Symmetry breaking is a crucial concept in cell polarity [37] referring to the process whereby a system transitions from a symmetric state to a polarized one. Symmetry can be broken by inherited cues or landmarks that identify a particular location in the cell, which in turn propagates to downstream protein

localization. Alternatively, polarity can arise by spontaneous symmetry breaking, in which a suitable network of interactions causes a system of proteins to self-assemble into an asymmetric pattern. Mechanisms for spontaneous symmetry breaking usually feature at their core a positive feedback, the classical example being the accumulation of Cdc42 during bud site selection in *S. cerevisiae* in the absence of Rsr1 [38]. Positive feedback amplifies a small, initially localized, fluctuation to the scale of the whole cell. This feedback can be generated through different network architectures and additional regulatory interactions may enhance the robustness of polarity [7]. The cooperative polar accumulation of RomR in the absence of MglA and MglB generates an effective positive feedback, raising the question of whether polar protein localization has its origin in spontaneous symmetry breaking by RomR. However, our data provides two lines of evidence against this. First, if RomR self-recruitment were responsible for symmetry breaking, we would expect *de novo* synthesized RomR in the absence of MglA and MglB to choose a polarity direction at random; instead, we found that the old cell pole was systematically favored. Second, systems that break symmetry by cooperative recruitment usually exhibit a characteristic bifurcation structure where the system is symmetric below a threshold protein concentration, beyond which asymmetry rapidly sets in; instead, we observe asymmetric RomR polar localization at all concentrations. Thus, these experiments suggest that rather than spontaneous symmetry breaking, RomR polar asymmetry in the absence of MglA and MglB is likely due to an unknown polar landmark that is inherited predominantly at the old pole during cell division. Importantly, the old pole preference is not observed in WT cells, although it remains unclear how MglA/MglB and/or their interactions with RomR nullify this preexisting bias.

In our mathematical model of the polarity protein interaction network, the generation of polarity by spontaneous symmetry breaking (i.e. without an old pole bias) emerges from the interplay between the RomR-MglB mutual recruitment and negative regulation of this feedback by MglA and occurs only in the presence of all three proteins. However, the strength of the latter regulation must be appropriately selected (see S9A Fig). If it is too weak, RomR and MglB will recruit one another effectively at both poles. Conversely, if it is too strong, accumulation of RomR and MglB will be suppressed at both poles. Only in an intermediate range of regulatory strengths can polar differentiation be sustained. Negative regulation of MglA by MglB is also required, and must be sufficiently strong as to prevent MglA from suppressing RomR-MglB accumulation at both poles (see S9B Fig). Conversely, our data suggest that direct negative regulation of MglB by MglA plays only a minor role in determining the wild-type asymmetry (see also S1 Text).

A key feature of the polarity module is that polarity can be inverted in response to Frz signaling. Thus, the polarity system must balance responsiveness to this signal against stability once polarity is established. Frz signaling is mediated by FrzX at the lagging pole and FrzZ at the leading pole [31, 32]. Guzzo et al [31] proposed that FrzX enables MglA to induce dissociation of MglB, although there is no direct evidence for this interaction. Our results are agnostic as to this mechanism, but suggest that direct regulation of MglB by MglA does not play a major role during the stable polarized phase. Within our interaction network, we can imagine other plausible points of action for FrzX/Z. Crucial to achieving an inversion of polarity is to establish a significant pool of MglA at the former lagging pole. Such a change could be instigated by FrzX locally downregulating the GAP activity of MglB and, thereby slowing the exclusion of MglA from this pole. However, this mechanism would reduce energy release through GTP hydrolysis, suggesting that an alternative energy source would be required to drive protein relocation. A similar effect of allowing for MglA accumulation at the lagging pole could be achieved by FrzX enhancing the recruitment of MglA by RomR/RomX, or enhancing the dissociation of MglB. At the same time, MglB and RomR must relocate to the former leading

pole. In part, this will inevitably occur as MglA accumulates at the old lagging pole, thereby inhibiting the MglB/RomR mutual recruitment at this pole. This effect could be enhanced if FrzZ at the former leading pole locally enhanced the recruitment or stability of MglB, or suppressed the negative effect of MglA on MglB/RomR accumulation at this pole.

Notably, while the model of Guzzo et al. [31] transitions from stable polarity into a relaxation oscillator upon constant Frz activation, our model showed no evidence of oscillatory dynamics, even for relatively large parameter variations. Ultimately, this is because both MglA and MglB localization depend on RomR, such that there is no clear separation between relocation timescales of the different proteins. In our model, rather than an oscillator, dynamic polarity in *M. xanthus* is akin to a spatial toggle switch, with stable polarized phases between discrete Frz-induced switching events. Notably, the *M. xanthus* polarity system appears to be capable of true toggling behavior, whereby the same signal (Frz activation) causes the state of the system (direction of polarity) to be inverted, regardless of the current state. This is in contrast to most so-called "genetic toggle switches" [39], in which distinct signals are required to shift the system out of each stable state. Rather, the spatially-extended nature of the system can be exploited so that the localized activities of FrzX/Z effectively modulate the Frz input according to the current polarity configuration, thereby achieving the kind of adaptive signaling required for true toggling behavior [40].

Rho GTPases are key regulators of polarity in eukaryotes. In many eukaryotic systems GEFs and GAPs localize to distinctive cellular locations to regulate GTPase activity in a spatially confined manner. For example, in the course of epithelial cell invagination during *Drosophila* embryogenesis, Rho1 colocalizes with its two cognate GEFs at the apical membrane while the GAP localizes to the basolateral membrane [41]. Similarly, in the one-cell *Caenorhabditis elegans* embryo RhoA and its cognate GEF colocalize at the anterior pole while its cognate GAP is at the posterior pole [42]. These systems usually combine positive feedback in Rho localization with mutual inhibition between polarized domains [7, 43]. A similar localization pattern and set of interactions, namely positive feedback to colocalize RomR/RomX and MglA at the leading pole combined with mutual inhibition with MglB, could equally generate polarity in *M. xanthus*. However, this is not the observed arrangement; rather, the GEF (RomR/RomX) and GAP (MglB) are primarily colocalized at the pole opposite to that of MglA, an arrangement that appears to promote an energetically-costly futile cycle of GTP exchange and hydrolysis at the lagging pole. Importantly, however, the above eukaryotic systems typically maintain stable polarity on long timescales, whereas polarity in *M. xanthus* is highly dynamic. Considering switching, allows us to rationalize the *M. xanthus* configuration. We propose that in this arrangement, GEF and GAP activities are present at both poles but coordinated so as to favor accumulation of MglA-GTP at the leading pole but not at the lagging pole. Upon Frz activation, preexisting GEF activity at the lagging pole allows MglA-GTP to rapidly accumulate here, in preference to the former leading pole. Such rapid MglA-GTP accumulation would be difficult if GEF (RomR/RomX) and GTPase (MglA) stably colocalized at the leading pole with GAP (MglB) at the lagging pole, thereby limiting the capacity of the system to switch polarity. Thus, we hypothesize that the *M. xanthus* polarity scheme reflects a trade-off between maintaining stable polarity and the capacity for Frz-induced polarity inversion. Interestingly, it has been shown that in some cases, eukaryotic Rho GEFs and GAPs also colocalize in regulatory complexes [44], in particular in excitatory synapses that undergo rapid remodeling in response to extracellular stimuli [45]. Rho GTPases are also important in the regulation of the dynamic polarity of migrating *Dictyostelium* and leukocyte cells [5], where the polarity direction can rapidly respond to extracellular cues. Furthermore, it been shown that both polar and oscillatory morphologies of melanoma cells can result from a combination of mutual antagonism between Rho and Rac GTPases and positive feedback mediated by mechanical interactions

with the extracellular medium [46, 47], which play an analogous role to that we propose for RomR. Therefore, similar principles to those which we have uncovered here may also be broadly applicable in achieving stable but responsive regulation of polarity in other cell types.

While we have identified the network topology for establishing and maintaining front-rear polarity in *M. xanthus*, several interactions remain to be characterized experimentally. For instance, it remains an open question how polar localization of RomR is brought about in the absence of MglA and MglB. The mechanism underlying the old-pole preference of RomR, and how this preference is nullified in the presence of MglA and MglB, also remain to be investigated. The molecular details underlying the positive feedback between RomR and MglB for polar accumulation are not understood and, along the same lines, it remains unclear how MglA may interrupt this positive feedback. Similarly, while our model rationalizes the identical localization patterns of MglB and RomR, it remains to be clarified how RomR/RomX GEF activity dominates over MglB GAP activity at the leading pole and MglB GAP activity dominates over RomR/RomX GEF activity at the lagging pole. Addressing these questions will be exciting areas for future investigations.

## Materials and methods

### Cell growth and construction of strains

All strains are derivatives of the *M. xanthus* DK1622 WT strain and are listed in S6 Table. Plasmids and oligonucleotides used are listed in S7 Table and S8 Table, respectively. *M. xanthus* was grown at 32˚C in 1% CTT broth [48] or on 1.5% agar supplemented with 1% CTT and kanamycin (50μg/ml) or oxytetracycline (10μg/ml) as appropriate. In-frame deletions were generated as described [49]. Plasmids were introduced in *M. xanthus* by electroporation and integrated by homologous recombination at the endogenous locus or at the *mxan18-19* locus. All in-frame deletions and plasmid integrations were verified by PCR. Plasmids were propagated in *Escherichia coli* TOP10 (F⁻, *mcrA*, Δ(*mrr-hsd*RMS-*mcr*BC), φ80*lacZ* ΔM15, Δ*lac*X74, *deo*R, *rec*A1, *ara*D139, Δ(*ara-leu*)7679, *gal*U, *gal*K, *rps*L, *end*A1, *nup*G). *E. coli* was grown in LB or on plates containing LB supplemented with 1.5% agar at 37˚C with added antibiotics when appropriate [50]. All DNA fragments generated by PCR were verified by sequencing.

### Immunoblot analysis

Cells were treated as for fluorescence microscopy in the presence or absence of 300 μM vanillate. At each time point, cells were collected from an agarose pad and samples prepared for immunoblot analysis. Proteins were separated by SDS-PAGE as described [50]. Immunoblot analysis was done as described [50]. Rabbit polyclonal α-MglA [22], α-MglB [22], α-RomR [29] and α-PilC [34] antibodies were used together with goat α-Rabbit IgG (whole molecule)-Peroxidase antibody (Sigma) as secondary antibody. The same membrane was probed with α-PilC antibodies as a loading control. Blots were developed using Luminata Crescendo Western HRP Substrate (Millipore) and quantified using a LAS-4000 luminescence image analyzer (Fujifilm).

### Cell imaging

For fluorescence microscopy, exponentially growing cells were placed on slides containing a thin pad of 1% SeaKem LE agarose (Cambrex) with TPM buffer (10mM Tris-HCl pH 7.6, 1mM KH$_2$PO$_4$ pH 7.6, 8mM MgSO$_4$) and 0.2% CTT medium, and covered with a coverslip. After 30 min at 32˚C, cells were visualized using a temperature-controlled Leica DMi8 inverted microscope and phase contrast and fluorescence images acquired using a Hamamatsu

ORCA-flash V2 Digital CMOS camera. For time-lapse recordings, cells were imaged for 6 hrs using the same conditions. To induce expression of genes from the vanillate inducible promoter [51], cells were treated as described in the presence of 300 μM vanillate. The data sets used for fluorescence microscopy quantification are available in S9 Table.

## Image analysis

Microscope images were first processed with Fiji [52] and cell masks first determined using Oufti [53] and manually corrected when necessary. Fluorescence was quantified in MATLAB (Mathworks) using custom scripts. Briefly, background fluorescence was determined by fitting a two-component Gaussian mixture model to the pixel intensities of all pixels in an image that were not within any cell mask. The background intensity was taken to be the mean of the Gaussian component with the greatest weight; typically, this component accounted for >90% of the pixels in the image. This background level was subtracted from all pixels. The total fluorescence of each cell was quantified as the sum of all background-corrected pixel intensities within the cell mask. For spot detection, the background-corrected fluorescence image was first filtered by convolution with a negative Laplacian of Gaussian (LoG) kernel with the form

$$K(i,j) = \frac{2\sigma^2 - (i^2 + j^2)}{2\pi\sigma^6} \exp\left(-\frac{i^2 + j^2}{2\sigma^2}\right),$$

where $i$ and $j$ are the distances from the center of the convolution kernel in the x- and y-directions. The kernel size ($L = 9$) and width parameter ($\sigma = 1.75$) were chosen to match the detected polar spots with those identified by inspection. This filter enhances spot-like features of the image while compressing the range of pixel intensities in non-spot regions. To avoid double counting polar spots from other nearby cells, pixels that were contained within other cell masks were set to zero prior to processing. To identify polar clusters, we constructed circular search regions at each pole with a radius of 10 pixels, centered on the fifth segment of the cell mask from the corresponding cell pole. This search region was chosen to extend slightly outside the cell mask as the masks often did not contain the entirety of polar fluorescence clusters. Within each search region, we identified pixels in the LoG-filtered image with intensity greater than a threshold of three standard deviations above the mean of all pixels within the cell mask but outside the two polar search regions. A pole was considered to have a polar spot if a contiguous set of at least three pixels above the threshold intensity was found within the corresponding polar search region. If more than one such set of pixels was detected within a given search region, the polar spot was taken to consist of the largest set of pixels. The polar fluorescence was quantified as the sum of pixel intensities of the pixels in the unfiltered image within the polar spot if any such spot was detected, or zero if there was no such spot detected. Since this method was less reliable in the relatively noisy imaging conditions of the induction experiments, these data were subsequently manually curated to remove false positive spot detections.

## Cell tracking and pole identity

Tracking of cell identities in movies was partially automated using a custom MATLAB script. Briefly, we examine the positions of cell poles in adjacent frames. For each cell mask in a given frame, the distances from the cell poles to the poles of each cell mask in the previous frame were calculated. If the total distance to the closest cell in the previous frame was lower than a threshold of 40 pixels, it is assumed that the mask represents the same cell. It is therefore assigned the same cell id as the matching mask in the previous frame. If the total distance was greater than this threshold, but the distance from one cell pole to the closest pole in the

previous frame was less than the threshold, then the cell was assumed to be a daughter of the corresponding cell in the previous frame. The pole that satisfied the distance criterion in the current cell was labeled as the "old" pole and the opposite pole was labeled as the "new" pole. If no matching pole was found in the previous frame (and for all cells in the first frame of the movie), then the mask was considered to correspond to a new cell, and no pole identity was assigned. The cell trajectories produced by this procedure were then inspected and manually corrected as necessary. In addition, trajectories that corresponded to the same cell, but that were marked as distinct because the cell was not detected in one intervening frame, were merged.

Tracking of motile cells during induction was first performed using Oufti, from which the cell outlines were obtained and then manually corrected. Direction of motility and leading/lagging pole determination was performed with a custom script written in MATLAB. Briefly, for every cell, the position in the XY plane of both poles, in every frame, was determined. Cell movement was considered when a cell moved at least 10% of its cell length, between consecutive frames, in order to avoid stochastic motions. Afterwards, the leading and lagging pole were determined based on the angle made between the line segment comprising the distance between both poles, and the line segment comprising the previous and new pole positions, between consecutive frames. Finally, fluorescence analysis was performed using the previously described method in Image Analysis.

### Description of mathematical model

Our model closely follows that of [31]. In particular, we retain the elegant structure of their model to describe polar protein localization patterns. For completeness we describe here the model in more detail as well as the model assumptions. The population of each of the three protein species (A, B and R, representing respectively MglA, MglB and the RomR/RomX complex) is divided between three cellular pools that represent the fraction of each protein that is localized at each of the two cell poles ($A_i$, $B_i$, $R_i$ for i = 1,2), and the delocalized fraction ($A_c$, $B_c$, $R_c$). The delocalized proteins are assumed to be well-mixed throughout the cytoplasm (see S1 Text). The rate of exchange of each of these proteins between the different pools is described by a set of ordinary differential equations, as shown in Fig 6B. Based on fluorescence recovery after photobleaching (FRAP) experiments [31], the exchange of proteins between the poles and the cytoplasm takes place on much faster timescales than protein translation and degradation. Therefore, these processes are neglected and the total amount of each protein is taken to be constant over time. Formally, therefore, the model is a 6-dimensional (9 protein pools—3 conservation laws = 6 degrees of freedom) non-linear system of differential equations.

The aim of our mathematical model was to test whether the interactions in Fig 6A are sufficient to explain the polarity pattern observed in snapshots of cells under steady-state conditions and, in particular, to explain how the WT pattern emerges from these interactions (Fig 5A, Fig 5B and Fig 5C), rather than to fully describe all details of the polarity system under all conditions. We, therefore, implemented the interactions in their simplest forms by choosing the lowest-order interactions, except for the direct interactions between polar MglA and MglB where we follow Guzzo et al. [31] in assuming a quadratic form since the active form of MglB is thought to be multimeric [28, 54]. The rationale for using the same form for both the dissociation of MglA by MglB and of MglB by MglA is the hypothesis that in a fraction of MglA GAP-induced dissociation events, MglB also dissociates as part of a MglB/MglA-GDP complex. We do not explicitly model MglA nucleotide exchange and GTPase activity, but instead these processes are included implicitly in the polar recruitment of MglA by RomR/RomX and exclusion by MglB. Protein deletion mutants were modeled by setting all pools of the

corresponding model component to zero. The old-pole bias in RomR localization was implemented by reducing the RomR dissociation rate $d_R$ by a constant factor at pole 1 only in all mutant conditions but not in the WT condition. It can be shown by direct solution and linear stability analysis that, in the absence of such a bias, the model equations (Fig 6B) permit only symmetric stable fixed points in any of the single- or double-mutant conditions, regardless of the choice of parameters (see S1 Text for details).

Different modes of action of MglA in WT were tested by setting $d_{BA} = 0$ or $K \to \infty$ (analysis of the regions of instability of each mechanism can be found in S1 Text). In the case of only direct regulation of MglB by MglA ($K \to \infty$) we were unable to find, either manually or by fitting, any combinations of parameters that resulted in spontaneous symmetry breaking in a regime consistent with the localization patterns observed in mutant conditions. Symmetry breaking was observed only in regimes in which MglA was largely polar in the absence of RomR (see S1 Text for details). Results in Fig 6C are for parameters in a regime consistent with mutant localization patterns (S5 Table).

Model parameters were chosen, through a combination of manual and automatic fitting, so as to closely match the polar fluorescence of the various deletion mutant strains and WT in steady state conditions (S5 Table; Fig 5A, Fig 5B and Fig 5C, see S1 Text for details). Briefly, the polar dissociation rates for each protein were fixed to be in accordance with according to the fluorescence recovery times measured in FRAP experiments [31]. With the dissociation rates fixed in this way, suitable values for the remaining model parameters (including the bias in RomR dissociation rates) were then chosen by using the hierarchy of double- and single-mutants to fix subsets of parameters where possible. Finally, the feedback parameters $d_{AB}$, $d_{BA}$ and/or $K$ were chosen by matching to the wild-type localization pattern. These manually-determined parameters were then used as the starting point for an automatic parameter fitting step, wherein we minimized the total squared deviation between experimental mean localization and model outputs,

$$\sum_{s \in strains} \sum_{X=A,B,R} \sum_{i=1,2} (X_{i,s,model} - <X_{i,s,exp}>)^2.$$

We found that without manual curation of the initial trial parameter values, global optimization was ineffective due to large regions of parameter space in which the model produces WT monostability.

## Simulations

Simulations were performed using a custom program written in C++ (available at https://github.com/gerland-group/mxanthus_polarity). In particular, the system of differential equations was integrated using the default Dormand-Prince 5<sup>th</sup>-order Runge-Kutta method of the Odeint library [55] from the Boost C++ library collection. Unless otherwise specified, all simulations were initialized with 1.1% of each protein at pole 1 and 1% at pole 2 and run for a simulation time of 1000 min, which was significantly longer than the time required to reach steady-state.

## Statistical analysis

Statistical details of each experiment including number and type of observations (n) and the corresponding dispersion measures can be found in either the Figures, Figure legends, or in S1–S4 Tables. In all cases, all quantified cells were included in the analysis and no outliers were removed.

Statistical significance of the difference in the polar localization patterns between strains was estimated with a two-dimensional two-sample Kolmogorov-Smirnov test using the method of Fasano and Franceschini [56]. This test estimates the likelihood of the null hypothesis that two sets of sampled data are drawn from the same underlying distribution, and does not rely on any assumptions about the underlying data.

Statistical significance of differences in the mean polar fluorescence and mean asymmetry between strains was determined using Welch's *t*-test. This test assumes that the sample mean is approximately normally-distributed, which following the central limit theorem will generally be the case for sufficiently large samples of non-normally-distributed data. In our data, n>100 for all strains. Validity of the resulting *p*-values was also estimated by bootstrapping with data resampling. Similar magnitudes of *p*-values were obtained from 1000 bootstrapping instances.

Statistical significance of old-new pole bias and leading-lagging pole bias was determined using binomial tests with a null model of no asymmetry (i.e. $p_{old} = p_{new} = 0.5$ or $p_{leading} = p_{lagging} = 0.5$). Since we have *a priori* no expectation of bias in either direction, a two-sided test was used. Thus, the quoted *p*-values represent the probability in the null model of obtaining a bias of at least the observed magnitude in favor of either pole. To test whether RomR-mCherry was significantly colocalized with PilQ-sfGFP, a one-sided binomial test was used with the null model of no bias in RomR-mCherry localization. Thus, the resulting *p*-value represents the probability of obtaining at least as many RomR-mCherry and PilQ-sfGFP cluster colocalizations in the null model as were observed. Binomial tests are assumption free.

Statistical significance of the difference between polarity inversion probabilities of different strains, and of difference in the old/new pole probabilities between strains, were determined using the $\chi^2$ test for independence, which estimates the likelihood of the observed probabilities arising under the null model that the true probabilities of the strains under comparison are the same. The $\chi^2$ test requires a sufficiently large number of occurrences of all the possible outcomes (typically, >5); in our data, the number of observations for all outcomes was >60.

In all cases, we take a *p*-value less than 0.05 to indicate statistical significance. All tests were performed using the corresponding functions in the SciPy Python library [57], except for the two-dimensional Kolmogorov-Smirnov test, which was reimplemented in Python following Press et al [58].

## Supporting information

**S1 Fig. Quantification of MglA-mVenus, MglB-mCherry and RomR-mCherry polar localization.** (A) MglA-mVenus localization in WT and in the absence of MglB and/or RomR. Cells were imaged by epifluorescence microscopy. Representative fluorescence images are shown together with *s*catter plots of the calculated fractions of fluorescence at pole 1 and pole 2 for all analyzed cells. Scale bar, 5 μm. In the scatter plots, the blue dots indicate the mean fractions at the two poles for a given strain. n, number of analyzed cells. (B) MglB-mCherry localization in WT and in the absence of MglA and/or RomR. Cells were imaged and analyzed as in A. (C) RomR-mCherry localization in WT and in the absence of MglA and/or MglB. Cells were imaged and analyzed as in A.
(EPS)

**S2 Fig. Quantification of MglA-mVenus, MglB-mCherry and RomR-mCherry polar localization in the absence of PilQ and/or AglZ.** (A, B, C) MglA-mVenus, MglB-mCherry and RomR-mCherry localization was determined in strains of the indicated genotypes. Cells were imaged and analyzed as in S1A Fig, S1B Fig and S1C Fig Scale bar, 5 μm.
(EPS)

**S3 Fig.** Quantification of MglA-mVenus, MglB-mCherry and RomR-mCherry polar localization in the absence of PilQ and/or AglZ (A, B and C) Polar localization of MglA-mVenus, MglB-mCherry and RomR-mCherry, respectively, in the absence of the other two proteins and in the absence of the motility machineries as indicated. First row, mean fraction of fluorescence at each pole for cells of indicated strains (filled circles). Dispersion of the single-cell measurements is represented by error bars and ellipses (dashed lines). Direction and length of error bars are defined by the eigenvectors and square root of the corresponding eigenvalues of the polar fraction covariance matrix for each strain. Color code for strains is indicated in row three. Second row, fraction of cells of each strain with two, one or no detectable polar clusters. Third row, mean localization for each strain (m.p.f., mean total polar fluorescence; ω, mean asymmetry; n, number of cells). Fourth and subsequent rows, histograms of the fraction of cells with a given total polar fluorescence.
(EPS)

**S4 Fig. RomR polar localization is stably maintained and correlates with the old pole in the absence of MglA and MglB.** (A) Tracking of RomR-mCherry localization in Δ*mglA* Δ*mglB* Δ*aglQ* mutant. Left panels, cells were imaged for 6 hrs with images captured every 10 min; cells labelled 1 and 2 are daughter cells following a division between 4:30 and 4:40 and cells labelled 3 and 4 are daughter cells following a division between 4:40 and 4:50. Right panels, polar fractions of RomR-mCherry in the cells labelled 1–4 in the images on the left following cell division. (B) Tracking of RomR-mCherry localization in the Δ*aglQ* mutant. Left and right panels as in A. (C) Autocorrelation of the polar fraction of RomR-mCherry in the Δ*mglA* Δ*mglB* Δ*aglQ* (grey dots) and Δ*aglQ* mutants (green dots). Polar fluorescence at each cell pole (*j*) at each frame *i* of the time-lapse recording, $p_i^{(j)}$, was normalized as $\Delta p_i^{(j)} = (p_i^{(j)} - \bar{p}_i)/\sigma_i$, where $\bar{p}_i$ and $\sigma_i$ are respectively the mean and standard deviation of the fractions of cell fluorescence associated with all cell poles in frame *i*. Autocorrelation functions were then calculated as $C(n\Delta t) = \langle \Delta p_i^{(j)} \Delta p_{i+n}^{(j)} \rangle_{(j),i}$, where *Δt* is the time interval between successive frames, and the average is taken over the polar fractions at all individual cell poles (*j*) and frames *i*.
(EPS)

**S5 Fig. Induction of *romR-mCherry*, *mglA-mVenus* and *mglB-mCherry*.** (A, B, C) Accumulation of fluorescent fusion proteins during induction analyzed by immunoblotting and by total fluorescence. Left panels, cells of the indicated genotypes were placed on thin pads containing 1% agarose buffered with TPM, 0.2% CTT medium and 300 μM vanillate. At each time point, cells were collected from an agarose pad and samples prepared for immunoblot analysis. Protein from the same number of cells were loaded per lane. Fusion proteins with their calculated molecular masses are indicated. Similarly, the predicted positions of the untagged proteins are indicated with their calculated molecular masses. In the strains in the leftmost lanes, genes for the fusion proteins were expressed from the relevant native site and were included here to compare vanillate induced protein levels to the levels obtained from the native promoter. The in-frame deletion mutants were included as negative controls. The PilC blots were included as loading controls. Middle panels, immunoblots were quantified and protein levels plotted as a function of time (orange) in % of the level in the strain expressing the relevant fusion protein from the native site. Average total fluorescence per cell (mean ± one standard deviation) was plotted as a function of induction time (black). Right panels, Fluorescence concentration (Methods) plotted as a function of induction time as mean ± one standard deviation. (D) Tracking of RomR-mCherry localization in Δ*mglA* Δ*mglB* Δ*romR* mutant during induction of *romR-mCherry*. Cells were treated as in A, imaged by fluorescence microscopy and polar fractions calculated. Shown are polar fractions for two representative cells as a

function of induction time.
(EPS)

**S6 Fig. PilQ-sfGFP has an old pole bias in the absence of MglA and MglB.** Cells of the indicated genotype were placed on thin pads containing 1% agarose buffered with TPM and 0.2% CTT medium, and then imaged for 6 hrs with images captured every 10 min. Cell division events were identified and in the daughter cells, the polar fractions of PilQ-sfGFP at the old and new pole quantified over time. Plotted are the mean ± one standard deviation of all observed cells at each time point. n: number of cells observed immediately after division. Because cells divide at different time points during the recording period, the number of cells included at each time point varies; however, at least 40 cells were included per time point. PilQ-sfGFP localization was significantly biased towards the old pole after cell division (two-sided binomial test, $p \ll 10^{-10}$).
(EPS)

**S7 Fig. RomR-mCherry localization correlates with the old pole in the absence of MglA or MglB.** Fraction of RomR-mCherry fluorescence at the old (orange) and new (blue) cell poles as a function of time after cell division in strains of the indicated genotypes. Both strains contain the Δ*aglQ* mutation to allow recording the same cells over extended periods of time. Plotted are the mean ± one standard deviation of all observed cells at each time point. n: number of cells observed immediately after division. Because cells divide at different time points during the recording period, the number of cells included at each time point varies; however, at least 16 cells were included per time point.
(EPS)

**S8 Fig. Exploring the dynamic establishment of polarity.** Simulated cells were initialized with polar asymmetry (1%) of two proteins, as indicated (left). For each of the initial arrangements shown, the system evolves to the same final state (right).
(EPS)

**S9 Fig. Parameter regions of spontaneous polarization.** (A) Bifurcation diagram showing the steady-state polar fractions as the strength of the negative feedback from MglA on RomR recruitment by MglB (*K*) was varied. In the symmetric regions, the fractions at the two poles coincide. (B) Phase-space plot of the steady-state RomR polarity, $\omega_R$, for different combinations of *K* and $d_{AB}$, the strength of negative regulation of MglA by MglB. All other parameters were fixed at their values in S5 Table. Dashed line indicates the estimated boundary between the polarized and symmetric regions.
(EPS)

**S1 Table. Summary of quantification of fluorescent fusion protein localization in different strains.**
(XLSX)

**S2 Table.** *P*-values for comparisons of polar localization distributions of fluorescent fusion proteins in different strains.
(XLSX)

**S3 Table.** *P*-values for comparisons of mean total polar fluorescence and mean asymmetry in different strains.
(XLSX)

**S4 Table. Summary of polarity inversion probabilities.**
(XLSX)

**S5 Table. Model parameters.**
(XLSX)

**S6 Table. Strains used in this study.**
(XLSX)

**S7 Table. Plasmids used in this study.**
(XLSX)

**S8 Table. Oligonucleotides used in this study.**
(XLSX)

**S9 Table. Data from fluorescence microscopy.**
(XLSX)

**S1 Text. Extended model discussion and analysis.**
(DOCX)

## Acknowledgments

We thank Anna Potapova for the gift of pAP37, Anke Treuner-Lange for the gift of pMAT123 and Sean Murray and Dobromir Szadkowski for many helpful discussions.

## Author Contributions

**Conceptualization:** Luís António Menezes Carreira, Filipe Tostevin, Ulrich Gerland, Lotte Søgaard-Andersen.

**Data curation:** Luís António Menezes Carreira, Filipe Tostevin.

**Formal analysis:** Luís António Menezes Carreira, Filipe Tostevin, Ulrich Gerland, Lotte Søgaard-Andersen.

**Funding acquisition:** Ulrich Gerland, Lotte Søgaard-Andersen.

**Investigation:** Luís António Menezes Carreira, Filipe Tostevin.

**Project administration:** Lotte Søgaard-Andersen.

**Resources:** Ulrich Gerland, Lotte Søgaard-Andersen.

**Software:** Filipe Tostevin.

**Supervision:** Ulrich Gerland, Lotte Søgaard-Andersen.

**Writing – original draft:** Luís António Menezes Carreira, Filipe Tostevin, Lotte Søgaard-Andersen.

**Writing – review & editing:** Luís António Menezes Carreira, Filipe Tostevin, Ulrich Gerland, Lotte Søgaard-Andersen.

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
