## [Decision Letter · Decision Letter 0]

14 Apr 2020

Dear Dr Søgaard-Andersen,

Thank you very much for submitting your interesting Research Article entitled 'Protein-protein interaction network controlling establishment and maintenance of switchable cell polarity' to PLOS Genetics. Your manuscript was fully evaluated at the editorial level and by independent peer reviewers. The reviewers appreciated the attention to an important topic but identified some minor aspects of the manuscript that should be improved.

We therefore ask you to modify the manuscript according to the review recommendations before we can consider your manuscript for acceptance. Your revisions should address the specific points made by reviewer 2.

[LINK]

Yours sincerely,

Daniel J. Lew

Guest Editor

PLOS Genetics

Gregory P. Copenhaver

Editor-in-Chief

PLOS Genetics

This study presents a logical and interesting set of experiments to characterize a bacterial cell polarity circuit. The experiments appear to be well executed and quantified, the inferred interaction topology is well supported, and the model provides additional insight with several quite simple assumptions. With the correction of the minor errors below, I recommend acceptance.

Fig. 2A,B-some labels on the bottom panels appear to be incorrect: A) mutant label should say deltaBdeltaR? and B) wild-type label should say A+R+?

Reviewer's Responses to Questions

**Comments to the Authors:**

Reviewer #1: This study presents a systematic analysis of the cell polarity underlying motility in M. xanthus. By studying polarity in fluorescence microscopy in all relevant strain backgrounds and doing a quantitative analysis (importantly distinguishing polar localization and asymmetry between poles as well as polar cluster intensity and fraction of cells with polar clusters), the authors very systematically dissect the regulatory networks underlying polarity. The results are then incorporated into a mathematical model to obtain further insight.

In my opinion, this is a exemplarily beautiful study and this is one of the rare cases where a paper could be accepted without revisions. The only point that remains open is the dynamics of switching, which in the “basic” scenario with only RomR does not happen, while it is typical for the WT. However, this is likely beyond the current study, but might be interesting to address in a follow-up study.

Reviewer #2: This is an interesting paper that addresses the polarity switching in a bacterium, combining experiments that quantify the abundance of singling molecules at the cell poles with a mathematical model to describe the dynamics.

Since my primary expertise is in eukaryotic cell polarity modelling, I have little to say about the specific biological details presented here, other than the comment that I found it interesting to see the similarities (as well as differences). I will mainly comment about the mathematical model and some suggestions for increasing clarity and readability.

Overall, I believe that the paper is of interest and merits publication.

Signed: Leah Edelstein-Keshet

Major Comments:

The authors are using a modified version of a model previously presented elsewhere, so this explains why the description of the model is brief. That said, the paper should be self-contained, so a little more detail would be useful, primarily in some text in a Supporting Information section. (See specific suggestions below). The authors mention some analysis (e.g. linear stability), and some of those details - even abbreviated - would be helpful in the Appendix. The first 2 equations of the model are a bit like species-competition equations, with R modulating the relative activation rates of the competitors. Such equations are relatively easy to analyze, and the parameter dependence can be easily visualized in bifurcation diagrams. In particular, the regimes of instability could be quantified fairly easily, I believe.

For the simulations, the authors used some custom programs, but since the ODEs are so very standard, they could be easily handled by commonly available software, e.g. XPP, MatLab, Morpheus. (In XPP bifurcation analysis would be easy to do. In Morpheus, the ODEs could also be extended to a bacterial-shaped elliptical domain.) An advantage of this is also that the code is easily shared with others or uploaded as as SI, so students can play with the model. It would make the paper much more fun to read and experience. I realize that some of this is perhaps a bit beyond the purpose of the current study (e.g. the spatial version), but I recommend it as a relatively easy way to go to the next step.

The models are all two-compartment ODEs (with conservation laws for the cytoplasmic compartment). This resembles the treatment of Rac-Rho polarity in melanoma cells in the pair of papers:

Holmes, William R., et al. "A mathematical model coupling polarity signaling to cell adhesion explains diverse cell migration patterns." PLoS computational biology 13.5 (2017): e1005524.

Park, JinSeok, et al. "Mechanochemical feedback underlies coexistence of qualitatively distinct cell polarity patterns within diverse cell populations." Proceedings of the National Academy of Sciences 114.28 (2017): E5750-E5759.

In that case, feedback from the extracellular matrix (ECM) affected the competition, replacing the role of the R in the model by the authors. One takeaway from that pair of papers was that it can be helpful to go through a couple of the "failed models" to show what is essential and what is not. (This could be done in the SI, and would make the paper more attractive to modellers.)

The authors say nothing about diffusion across the cell. It would be helpful to get an estimate of the diffusion times, and how they compare with the other kinetic timescales. I suspect that it is entirely reasonable to ignore diffusion in these small cells, but a brief reasoning in the SI is justified. I liked the description of parameter estimation and hope this could be detailed further in the SI, as it is an important aspect of the model realism.

Minor issues

Circles in Fig 1A: represent molecules as well as amounts.. possibly confusing. The caption does not help explain this, and the font size on the lower part of Fig1A is too small.

Add the definitions of A, B, R to some figs such as Figs 2, 5, 6

In Description of mathematical model. (L 745-759) you can easily add the definitions of the variables, e.g.

cellular pools that represent the fraction of each protein that is localized at each of the two cell poles, and the

delocalized fraction.

cellular pools that represent the fraction of each protein that is localized at each of the two cell poles (A_i, B_i, R_i for i=1,2) , and the delocalized fraction (A_c, B_c, R_c).

Also include this in Fig 6 caption for better readability, and define A,B, R as levels of given components. Explain in caption for E that the size of the balls represents the amount of given substance that is localized at a pole.

L795

With the dissociation rates are fixed in this way

When the dissociation rates are fixed in this way

**Have all data underlying the figures and results presented in the manuscript been provided?**

Reviewer #1: Yes

Reviewer #2: Yes

PLOS authors have the option to publish the peer review history of their article (what does this mean?). If published, this will include your full peer review and any attached files.

Reviewer #1: No

Reviewer #2: Yes: Leah Edelstein-Keshet

---

## [Editor Report · Decision Letter 1]

21 May 2020

Dear Dr Søgaard-Andersen,

We are pleased to inform you that your manuscript entitled "Protein-protein interaction network controlling establishment and maintenance of switchable cell polarity" has been editorially accepted for publication in PLOS Genetics. Congratulations!

Yours sincerely,

Daniel J. Lew

Guest Editor

PLOS Genetics

Gregory P. Copenhaver

Editor-in-Chief

PLOS Genetics

Comments from the reviewers (if applicable):

**Data Deposition**

http://datadryad.org/submit?journalID=pgenetics&manu=PGENETICS-D-20-00335R1

**Press Queries**

---

## [Editor Report · Acceptance letter]

15 Jun 2020

PGENETICS-D-20-00335R1 

Protein-protein interaction network controlling establishment and maintenance of switchable cell polarity 

Dear Dr Søgaard-Andersen, 

We are pleased to inform you that your manuscript entitled "Protein-protein interaction network controlling establishment and maintenance of switchable cell polarity" has been formally accepted for publication in PLOS Genetics! Your manuscript is now with our production department and you will be notified of the publication date in due course.

With kind regards,

Jason Norris

PLOS Genetics

On behalf of:
